# Tensiomyography Allows to Discriminate between Injured and Non-Injured Biceps Femoris Muscle

**DOI:** 10.3390/biology11050746

**Published:** 2022-05-13

**Authors:** Srđan Đorđević, Sergej Rozman, Petra Zupet, Milivoj Dopsaj, Nicola Maffulli

**Affiliations:** 1TMG-BMC, Štihova ulica 24, 1000 Ljubljana, Slovenia; sergej.rozman@tmg.si; 2Institute for Medicine and Sports, Cesta na Poljane 24, 1000 Ljubljana, Slovenia; petra.zupet@i-ms.si; 3Faculty of Sport and Physical Education, University of Belgrade, 11000 Belgrade, Serbia; milivoj.dopsaj@fsfv.bg.ac.rs; 4Department of Physical Education and Health, Institute of Sport, Tourism and Service, South Ural State University, Prospekt Lenina 76, 454080 Chelyabinsk, Russia; 5Centre for Sports and Exercise Medicine, Barts and the London School of Medicine and Dentistry, Mile End Hospital, 275 Bancroft Road, London E1 4DG, UK; n.maffulli@qmul.ac.uk; 6School of Pharmacy and Bioengineering, Keele University School of Medicine, Stoke-on-Trent ST4 7QB, UK; 7Department of Musculoskeletal Disorders, Faculty of Medicine and Surgery, University of Salerno, Via S. Allende, 84081 Baronissi, Italy

**Keywords:** biceps femoris injuries, tensiomyography, functional diagnostic, contraction time

## Abstract

**Simple Summary:**

The hamstring muscle group is the most frequently injured muscle group in non-contact muscle injuries involving high-speed running sports. The biceps femoris (BF) muscle has the highest injury incidence. Clinical assessments and magnetic resonance imaging (MRI) are routinely used to diagnose a given injury. We tested the possibilities of a new technology—tensiomyography—as a potential screening test (diagnostic and classification purposes) to assess the functional differences between injured and non-injured BF muscles. The results show that tensiomyography has a high predictive ability to discriminate between injured and non-injured BF non-invasively and functionally and that it can be reliably used as a complementary screening test in the diagnosis of BF injuries.

**Abstract:**

The hamstring muscle group is the most frequently injured muscle group in non-contact muscle injuries in sports involving high-speed running. A total of 84% of hamstring injuries affect the biceps femoris (BF) muscle. Clinical assessments and magnetic resonance imaging (MRI) are routinely used for diagnosis and plan management. MRI-negative scans for clinically diagnosed hamstring injuries range from 14% to 45%. We tested the hypothesis that the functional differences between injured and non-injured BF assessed by tensiomyography can be used for diagnostic and classification purposes. We compared an injured group of 53 international-level soccer players and sprinters with 53 non-injured international-level soccer players and sprinters of both sexes. Comparing the injured vs. non-injured athletes and the left vs. right side in all of the athletes, we used the percentage of absolute differences in the BF contraction time (Tc) to classify non-injured and injured BF muscles. The receiver operating characteristic (ROC) curve and the area under the curve (AUC) and the precision–recall curve (PRC) were used to measure the classification accuracy and to identify cut-off limits using the Tc differences. There was a very high ROC AUC value of 0.981 (SE = 0.009, *p* < 0.000), with 98.11% of the injured muscles being correctly classified (cut-off point 12.50% on Tc differences), and an AUPRC value of 0.981, with association classification criteria at >9.87. Tensiomyography has a high predictive ability to discriminate between injured and non-injured BF non-invasively and functionally.

## 1. Introduction

Top athletes have daily and even multiple daily training sessions that are conducted at different levels of intensity. In addition to training, they also participate in regular competitive events during the annual season. Because of this, elite sport performance is associated with a high risk of musculoskeletal soft tissue injury. Muscle injuries typically result in the inability of the muscle–tendon complex to resist counter-applied forces while the muscle is lengthening. Muscles, tendons, and ligaments are usually injured during eccentric contractions [1,2]. The overall injury rate of professional soccer players is approximately 1000 times higher than that of typical industrial occupations that are generally regarded as high risk [3]. Hamstring muscle injuries are the most common type of injury in sports in which fast running and repeated sprints are the basic movement patterns [4]. The biceps femoris (BF) is the most injured hamstring muscle [5,6,7,8,9], and the muscle–tendon junction and adjacent muscle fibers are the most common sites of disruption [8]. Typically, BF injuries occur in sprinting, jumping, kicking, and tackling during high-force and high-speed movement tasks [8]. The transition between non-injured and injured skeletal muscles is continuous and is only based on definitions that set arbitrary distinctions between the two. Different approaches are used when defining and diagnosing muscle injuries. In soccer, for instance, a “hamstring injury” is defined as “a traumatic distraction or overuse injury to the hamstring muscle leading to a player being unable to fully participate in training or match play” [6]. Electromyographic (EMG) evidence shows that the hamstrings are active from mid-swing until terminal stance [10].

Based on a consensus conference [2], four types of muscle injuries/disorders and six subtypes have been introduced, and, since 2002, determining the injury size has been based on imaging, and a continuous scale is used to try and associate it with clinical outcome [11]. In the case of structurally identified muscle injuries, the exact anatomical site involved and the severity at the imaging assessment were used to develop a simple classification system [12]. In 2014, a new anatomically focused muscle grading system graded injuries from 0 to 4 and divided them into the subcategories a, b, and c based on the type of the injury: “myofascial”, “musculotendinous”, or “intratendinous” [13].

However, there is limited evidence to support either the pathological or prognostic validity of clinical and imaging grading systems, and MRI does not add any additional predictive value for the time to return to sport compared to baseline patient history and clinical examinations alone after an acute hamstring injury. In approximately 20% of cases, MRI could not distinguish between an injured and non-injured hamstring [11,14].

Hamstring injuries are highly prevalent in sports and mostly occur in non-contact circumstances when the athlete is sprinting, and approximately 80% of hamstring injuries involve the BF [15,16,17]. A critical factor in managing such injuries and planning a return to sport or training strategies is the availability of diagnostic tools. Return to sport is a multistage process and can be divided, for example, into a diagnosis, rehabilitation, specific post-rehabilitation training, high-intensity training, and, ultimately, the decision procedure for return to play. The transition between the various stages needs to marry functional and imaging results carefully. In this respect, the functional diagnostic procedure described in the present investigation may bridge the existing gap. In the later stages of the healing process following a muscle injury, it is possible to measure voluntary force to evaluate functional changes (isometric knee flexion strength deficit) in isometric [18] or isokinetic conditions (deficit for hamstring concentric 60°/s, 300°/s and hamstring eccentric [19]) with reasonable risk, including selective measurements of the BF or hamstrings (similar to Askling’s H-test [20]).

Mechanomyograms investigate muscle function through a quantitative evaluation of the low-frequency transverse oscillations generated from active muscle fibers that travel to the skin’s surface during contraction [21,22,23,24,25,26,27,28]. Several types of transducers [23] can be used to record mechanical muscle activity: piezo-electric contact sensors, microphones, laser distance sensors, electret condenser microphones, optoelectronic position-sensitive detectors, and accelerometers. Different mechanical aspects of muscle contraction can be assessed [23], such as the gross lateral movement of the contracting muscle, the subsequent vibrations at the resonance frequency of the muscle, and the dimensional changes/displacement in the active muscle fibers. Tensiomyography (TMG), a type of mechanomyography [29,30,31], assesses muscle function by measuring the changes in muscle form during an electrically induced contraction. TMG uses a displacement sensor to detect the radial enlargement of the muscle belly during a contraction. TMG has been used to measure muscle contractile properties and allows twitch contraction properties, including twitch contraction time, delay time, and the amplitude of BF contractions, to be investigated [32,33,34,35,36,37]. The reliability and repeatability of TMG measurements have been tested for different contraction parameters (delay time, contraction time, displacement, etc.) and have been found to be highly reliable [38,39].

Functional and structural muscle disorders (e.g., injuries) result in functional changes in an injured muscle. TMG is able to measure and quantify the contractile properties of a muscle non-invasively, which is the main reason for using this method in the present study. The present study tested the null hypothesis of no differences being observed in the contraction properties of injured and non-injured BF muscles in elite athletes assessed by tensiomyography. The next step was to investigate the highest sensitivity and specificity of the TMG classifier to separate injured and non-injured BF muscles by applying the optimal cut-off values. Based on the comments above, it can be concluded that the primary aim of this study is to test the null hypothesis of there being no differences in the contraction properties of injured and non-injured BF muscles in elite athletes assessed by tensiomyography. The secondary aim was to investigate the highest sensitivity and specificity of the TMG classifier to distinguish between injured and non-injured BF muscles by applying the optimal cut-off values.

## 2. Materials and Methods

### 2.1. Subjects

The injured group was composed of 53 (22 females and 31 males, age—24.2 ± 4.6 years; body height, BH—180.4 ± 7.6 cm; body mass, BM—70.9 ± 8.4 kg; body mass index, BMI—21.72 ± 1.28 kg/m^2^; training experience —14.0 ± 5.2 years; amount of training 7.8 ± 1.2 training/week) international-level soccer players and track and field sprinters (100 and 200 m).

We included athletes with acute, sudden pain in the posterior thigh when training or competing. Clinical examinations, conducted within 24 h of the injury, revealed: (i) localized pain when palpating the hamstring muscles, (ii) localized pain when performing a passive straight leg raise test, and (iii) increased pain when adding an isometric hamstring contraction during a passive leg raise. Each subject underwent an imaging diagnostic, magnetic resonance imaging—MRI (Figure 1), or ultrasound (US). Tensomyographic measurements were performed after clinical examination within 12 and 72 h of the index injury. The athletes participated in no organized physical activity for at least 60 min before the TMG measurements. The measurements were always taken between 10 am and 4 pm in a temperature-controlled room. During the measurements, the temperature was between 21 and 25 °C.

The non-injured group (controls) consisted of 53 subjects (22 females and 31 males, age 23.4 ± 4.3 years; BH—181.9 ± 6.9 cm; BM—70.9 ± 7.0 kg; BMI—21.39 ± 0.83 kg/m^2^; training experience—12.6 ± 4.3 years; amount of training 7.4 ± 1.3 training/week) who were active in the same sport and at the same level as the injured participants and who did not report either a history of hamstring injuries for at least two years or a history of chronic low back pain. All of the female subjects in the injured and non-injured groups were sprinters. The researcher who performed the tensiomyographic measurements was not informed as to whether an individual was injured or not, and the athlete was asked to not tell the assessor whether he/she had been injured.

The inclusion criteria for the subjects in the study were as follows: the subjects had to be national or international-level adult athletes who played in the first team (soccer players) or who competed in the first team (track and field athletes) of their clubs. The exclusion criteria for the subjects were a hamstring injury within the last two years and a history of chronic low back pain.

All of the procedures performed in the present investigations were conducted following approved guidelines, and all protocols received Institutional Human Research Ethics approval. Informed consent was obtained from all of the respondents.

The study was approved by the Commission of the Republic of Slovenia for Medical Ethics, approval study number—No. 125/03/14, and by the Institutional Ethical Board of Faculty of Sport and Physical Education, University of Belgrade Serbia, study approval number—484-2.

### 2.2. Measurement Procedures

For the tensiomyographic measurements, an inductive sensor incorporating a spring with a spring constant of 0.17 N/mm was used. The sensor provided an initial pressure of approximately 1.5 × 10^−2^ N/mm^2^ on a tip area of 11.34 mm^2^. A single-twitch electrical stimulus (a DC pulse of 1 ms duration) induced an isometric muscle contraction. Both electrodes (UltraStim^®^ Wire, Axelgaard, Fallbrook, CA, USA) were placed symmetrically 2 cm away from the sensor; the positive electrode (anode) was placed proximally, and the negative electrode was placed (cathode) distally [33,34]. The electrodes were self-adhesive. The measured muscle responses were stored and analyzed using a standardized algorithm for the TMGTM S1 system (TMG-BMC, 1000 Ljubljana, Slovenia) to determine the muscle response.

Three parameters were determined to evaluate the TMG signal: Td (delay time), Tc (contraction time), and Dm (displacement of the muscle belly during contraction). We only used highly reproducible parameters that were determined based on previous reliability studies [40,41]. A typical TMG record, with all of the parameters defined, is shown in Figure 2.

The duration of the total measurement procedure for both legs in a single subject was between 4 and 6 min. No subject reported any discomfort during or after the procedure.

### 2.3. Experimental Design

Subjects lay face down on a bed, and BF measurements were taken for both legs under isometric conditions. The knee was flexed to 15 degrees, and the sensor was placed on the BF at the midpoint of the line between the fibula head and the ischial tuberosity (Figure 1). After positioning the electrodes, electrical stimuli were introduced and repeated at 5 s intervals. The current was increased in 5 mA steps (stimulus duration of 1 millisecond) until a supramaximal muscle response was reached. If stimulation caused pain in the stimulated area, the measurement was immediately discontinued and was not considered in further analysis.

### 2.4. Variables

The following variables and abbreviations were used in the present study:

Tc—contraction time, expressed in ms; Td—delay time, expressed in ms; Dm—displacement (Figure 2), expressed in mm; BF—m. biceps femoris; diff—the percentage of differences between different BF; f—female subjects; m—male subjects; iN injured subjects; Ni—non-injured subjects; in—injured BFs; ni—non-injured BFs; L—left BF; R—right BF.

Example of combinations:

Tc L ni = Tc of left BF of non-injured subjects;

fTc L ni = Tc of left BF of female non-injured subjects;

mTd D in= Td of male injured subjects;

Tc iN ni = Tc of non-injured BF in injured subject group;

Tc diff = percentage of the BF Tc absolute difference;

Tc diff in = percentage of the absolute difference between injured and non-injured BF Tc;

Tc diff ni = percentage of the absolute difference between non-injured left and right BF Tc;

fTc diff ni = percentage of the absolute difference between non-injured left and right BF

Tc in the female subgroup;

mTc dif in = percentage of the absolute difference between injured and non-injured BF

Tc in the male subgroup;

The same classifications were also used for Td and Dm.

AUCf = ROC area under the curve for Tc classifier in the female subgroup;

AUCm = ROC area under the curve for Tc classifier in the male subgroup.

### 2.5. Statistical Analysis

All of the descriptive data are expressed as the mean ± SD. The Student’s t-test for paired samples was used to analyze the differences between the injured and non-injured muscles. The level of statistical significance was set at *p* < 0.05. Receiver operating characteristic (ROC) analysis was used to evaluate the discriminatory power of the tensiomyographic diagnostic test for BF injury detection. The ROC curve is a performance measurement that classifies the strength of various thresholds settings. The ROC curve is a probability curve, and the AUC curve represents the measure of separability. It describes how much the model is capable of differentiating between classes. The method proposed by Hanley and McNeil [42] and DeLong et al. [43] was used to compare the differences between two independent ROC curves with the same classifier.

The criteria for ROC analysis (classifiers) were the percentage of the absolute difference in the TMG parameters (Tc, Td, and Dm) between the injured and non-injured BF muscles in the injured group and the percentage of the absolute difference in the Tc between the left and right BF muscles in the non-injured group. We calculated the Youden index J. J indicates the maximum performance of a given cut-off when the sensitivity and specificity reach a maximum score and when they are equally important diagnostically. Additionally, the precision–recall curve (PRC) was used to define the relationship between precision (=positive predictive value) and recall (=sensitivity) for every possible cut-off at the explored variables [44]. A Monte Carlo simulation is a model used to predict the probability of different outcomes when the intervention of random variables is present. In addition, we used Monte Carlo simulation to compare the prospective and post hoc power functions. In all cases, we set a two-sided alpha at α = 0.05 and Monte Carlo sample size at 1000 [45].

The Kruskal–Wallis ANOVA test was used to compare non-normal distributions, and the Kolmogorov–Smirnov normality test was used to compare the distributions. MedCalc^®^ Version 18.8.1 (MedCalc Software, 1993–2014, Ostend, Belgium) and IBM SPSS Statistics 23 (IBM Corporation, Armonk, NY, USA) were used for additional statistical analysis. The sample size and power analysis were calculated using the G*Power 3.1.9.4 statistical software (Franc Faul, University of Kiel, Kiel, Germany, ©1992–2019). The results show that for the two groups and a total sample size of 106, the effect size d for 0.2—small, 05—medium, and 0.8—large power was 0.175, 0.722, and 0.983, respectively.

The percentage of absolute differences was defined as (Equation (1)).
(1)$$\mathrm{Tc}\left(\mathrm{diffiN}\right)=\frac{\frac{\mathrm{ABS}\left(\mathrm{Tcin}-\mathrm{Tcni}\right)}{\mathrm{Tcni}}*100+\frac{\mathrm{ABS}\left(\mathrm{Tcni}-\mathrm{Tcin}\right)}{\mathrm{Tcin}}*100}{2}$$
where Tc (diffiN) is the average of the absolute difference between the contraction time of the injured and non-injured BF, Tcin is the contraction time of the injured BF, and Tcni is the contraction time of the non–injured BF, all from the same subject in the injured group. The same calculation was used for Td and Dm.

For the negative control, we used the absolute percentage of the difference between the left and right Tc from the same subject in the non-injured group (Equation (2)).
(2)$$\mathrm{Tc}\left(\mathrm{diffNi}\right)=\frac{\frac{\mathrm{ABS}\left(\mathrm{Tcl}-\mathrm{Tcd}\right)}{\mathrm{Tcd}}*100+\frac{\mathrm{ABS}\left(\mathrm{Tcd}-\mathrm{Tcl}\right)}{\mathrm{Tcl}}*100}{2}$$
where Tc (diffNi) is the average of the absolute difference between the contraction time of the left and right BF, Tcl is the contraction time of the left BF, and Tcr is the contraction time of the right BF, all of which were from the same subject in the non-injured group.

All of the statistical analyses were performed separately on the female and male subjects in the injured and non-injured groups.

## 3. Results

The descriptive statistical data for all of the subjects (both injured and non-injured) are presented in Table 1 as the mean ± SD. The Kolmogorov–Smirnov normality test was used to examine whether the variables are normally distributed. To the compare Tc, Td, and Dm parameters, the two-tailed *t*-test was used, and significance was set at *p* = 0.05. Injured *N* = 53 (female 22) and non-injured *N* = 53 (female 22). No statistically significant differences were found when comparing the BF contraction properties between males and females, except when comparing non-injured fDm L and fDm vs. mDm L and mDm D. A statistically significant difference was found when comparing Td in, Tc, and Dm in vs. Td ni, Tc ni, and Dm ni.

No evidence of a statistically significant difference in the Dm was observed in any of the injured (4.9 ± 1.8 mm) and non-injured (5.0 ± 1.9 mm) BF muscles (Table 1, *p*-value = 0.683). All of the non-injured subjects showed no evidence of a statistically significant difference when comparing the left and right side of the BF Td (Td L = 22.8 ± 2.4 ms and Td R = 22.8 ± 2.1 ms), Tc (Tc L = 24.5 ± 4.7 ms and Tc R = 24.5 ± 4.5 ms), or Dm (Dm L = 4.6 ± 1.8 mm and Dm R = 4.8 ± 1.7 mm, Table 1). Considering the results of female and male subjects in the injured and non-injured groups, no statistically significant differences were observed between the contractile properties of the BF in the non-injured female and male subjects, with the exception of Dm female (Table 1, non-injured group female vs. male Dm, fDm-L-ni = 4.1 ± 1.3 mm and mDm-L-ni = 5.0 ± 2.0 mm, *p* = 0.049 and fDm-R-ni = 4.2 ± 1.2 mm and mDm-R-ni = 5.1 ± 1.9 mm, *p* = 0.035).

Figure 3 shows a typical example of injured and non-injured BF signals from the same subject. Comparing the Tc and Td of the injured subjects in the injured and non-injured BF muscles, statistically significant differences were found for Tc (all injured 32.9 ± 8.5 ms vs. all non-injured 24.6 ± 5.1 ms; *p* < 0.001) and for Td IN (all injured 25.0 ± 3.6 ms vs. all non-injured 23.1 ± 2.3 ms; *p* < 0.001, Figure 4).

The absolute percentage of the differences between the injured and non-injured BF Tc and Td in the injured subjects and absolute percentage of the differences between the left and right leg in non-injured BF subjects are presented in Figure 4 and Figure 5, respectively.

The absolute percentage of the differences between the injured and non-injured BF Tc, Td, and Dm in the injured subjects was compared to the absolute percentage of the differences in the left and right BF Tc, Td, and Dm of a non-injured subjects and applied for injury classification. The results of the absolute difference in percentages are presented in Table 2. 

In addition, the results of the Monte Carlo simulations as a model for predicting the probability of different outcomes of the intervention of random variables show that there was no statistically significant difference in the delta values of Td and Dm between the nominal variables of injured and non-injured subjects (Cramer’s V value = 0.991, Approx. Sig. = 0.346, and Monte Carlo Sig. = 0.102 and Cramer’s V value = 0.991, Approx. Sig. = 0.399, and Monte Carlo Sig. = 0.313, respectively), while a statistically significant difference was found in the variable delta value Tc (Cramer’s V value = 1.000, Approx. Sig. = 0.297, and Monte Carlo Sig. = 0.014).

Using a receiver operating characteristic curve (ROC) for diagnostic accuracy and classification (Figure 6 and Figure 7), the absolute percentages of the differences in the contraction time (TcdiffiN) and (TcdiffNi) in all of the subjects showed a very high AUC value 0.981 (standard error = 0.009; the 95% confidence interval ranges from 0.934 to 0.998, and the significance level *p* = 0.000). The complete results of the ROC curve analysis for the measured TMG parameters are shown in Table 3.

The results of the ROC curve analysis show that the Tc classifier had the highest AUC, sensitivity, and specificity for the different cut-offs compared to Td and Dm (Table 3). Additionally, the absolute (summarized F + M) differences in Td and Dm are much less accurate in distinguishing between injured and non-injured BF muscles (AUC Td—0.712 and AUC Dm—0.665) than Tc (AUC Tc—0.981). Additionally, the Youden index for Tc was J = 0.868 (J provides a criterion for choosing the “optimal” threshold value regarding specificity and sensitivity) based on the maximum sensitivity (98.1%), specificity (88.68%), and cut-off point scores at differences of 12.50% (see Figure 8). Therefore, a TcdiffiN value of 12.50% results in an excellent 98.11% true positive ratio of correctly classified injured BF muscles and a false positive ratio of 11.32% (Figure 7). The false negative ratio (the ratio of injured subjects whose test results were wrongly classified suggesting that they are non-injured compared to all of the subjects who are injured) was low (<1.89%) when using the same cut-off limit. If perfect sensitivity is preferred, then the cut-off value of 9.89% (TcdiffiN) would obtain 100% (0 false negatives) sensitivity and a 13.21% false positive ratio. The cut-off limits range from 9.88 to 14.22% of the absolute difference between contraction times in the injured and non-injured BF muscles (Figure 7).

For Td, the AUC was 0.712 at the significance level of *p* = 0.000 and had a standard error of 0.052. The 95% confidence interval ranged from 0.62 to 0.80. For Dm, the AUC was 0.665 at the significance level of *p* = 0.003 and had a standard error of 0.055, and the 95% confidence interval ranged from 0.57 to 0.75. The results of the actual study show that the Tc classifiers (TcdiffiN) and (TcdiffNi) were markedly better than both Td and Dm (Table 3, AUC = 0.981, and the significance level *p* = 0.000). Additionally, it is important to emphasize that applying an ROC analysis in relation to injured and uninjured subjects and using the Tc absolute differences made it possible to classify injured from non-injured BF successfully, regardless of gender. For the female subjects, the AUC_f_ value was 0.969 for an optimal cut-off value at 12.50% of the Tc differences. For the male subjects, the AUC_m_ was 0.989 for an optimal cut-off value at 13.57% of the Tc differences between the injured and non-injured BF muscles (Figure 9). Using the same classifier, no statistically significant differences were observed between these two ROC curves (difference −0.02, SE = 0.0223, z statistic = −0.897, and significance level *p* = 0.369).

The results of the PRC curve analysis show very similar results to the results of the ROC curve analysis. The Tc classifier had the highest AUPRC, which ranged from 0.970 to 0.988 for the female and male subjects, respectively (Table 4). Additionally, the Tc positive predictive value (PPV) as a measure of precision and the true positive rate (TPR) as a measure of sensitivity showed the (Females, Males and Summarized F + M) highest classification level between the injured and non-injured BF groups (from 0.84, i.e., 84% for TPR to 1.00, i.e., 100% for PPV regardless of the subsample). It is important to emphasize that the PRC analysis with AUPRC, F1_max_, PPV, and TPR showed that the analyzed experimental data are balanced and can be used as a valid binary data classification method (all values are highly above the 0.5 threshold level).

## 4. Discussion

The results of the present study show that tensiomyography allowed different contraction properties to be identified and quantified between injured and non-injured BF muscles in elite athletes at Tc diff variables. The investigation quantified the differences in the twitch contraction properties of injured and non-injured BF muscles and used these differences as a classification test between the injured and non-injured muscles. On the selected cut-off values, sensitivity and specificity were evaluated using a receiver operating characteristic (ROC) analysis (Figure 9, Tc diff variables AUC_f_ = 0.969, cut-off = 12.50% and AUC_m_ = 0.989, cut-off = 13.57% for females and males, respectively). Additionally, the results of the PRC curve analysis show almost similar results to the ROC curve analysis, where it was shown that the Tc classifier had the highest AUPRC, which ranged from 0.970 to 0.988 for the female and male subjects, respectively (Table 4). The Tc positive predictive value (PPV), as a measure of precision, and the true positive rate (TPR), as a measure of sensitivity, showed the highest classification level between the injured and non-injured BF groups (from 0.84, i.e., 84% for TPR to 1.00, i.e., 100% for PPV regardless of the subsample). In addition, discriminative classification potential for the Tc diff variable was additionally confirmed by the application of Monte Carlo analysis, where only at Tc diff variable statistical differences between injured and non-injured subject samples were found (Cramer’s V value = 1.000, Approx. Sig. = 0.297, Monte Carlo Sig. = 0.014).

This study showed that tensiomyography is a functional assessment method that allows the functional changes in the muscle under examination after an injury to be assessed and quantified. Functional and structural muscle disorders (injuries) result in functional changes in the injured muscle. Indeed, functional (not structural) muscle disorders cannot be detected with current medical imaging techniques. The ability of TMG to measure and quantify the contractile properties of the muscle non-invasively was the main reason for using this method in the present study. Tensiomyography is non-invasive and does not involve force or torque measurements. Finally, the procedure is well-tolerated, relatively short, and easily repeatable, allowing a single muscle to be assessed within a few minutes. In this way, repeated assessments of large groups are possible, allowing clinically relevant longitudinal data to be collected.

Functional quantitative diagnostic procedures are lacking in the early phases of the recovery period. Current data suggest [46] that the earliest events associated with injury are mechanical and are reflected in the mechanical properties of the muscle. After an injury, the contractile properties of the muscle change [46,47]. Consequently, assessing these changes could be a way to monitor an injury and subsequent recovery.

Sprinters and soccer players are a group of athletes who have the added risk of hamstring injuries and, in particular, biceps femoris injuries [48]. They represent a relatively large group of athletes and are suitable for research. Female sprinters (at the international level) have similar maximum sprint running speeds to soccer players (at the national team level), which means they have a comparable relative load level on the BF during intense sprinting. Factors influencing the increased probability of BF injuries in these groups may be similar (maximum or near-to-maximum running speed [49]) or different related to the specificity of the movement pattern (kicking [49]) and factors related to neuromuscular fatigue (intermittent type of loading in soccer) [50,51].

Our study found no statistically significant differences in BF contractile properties between male and female populations of healthy and injured muscles. Based on these findings, it can be concluded that female athletes do not influence final results differently than other subjects.

In the present study, changes in the activation patterns were observed in the injured muscle. In a muscle injury, preferential damage occurs to type II muscle fibers [52,53]. This, together with indications that human fast hybrid fibers (IIa/IIx MHC isoform) are more sensitive to standardized in vitro eccentric contractions [54], can be used to better explain the connection between a longer Tc and Td and muscle injury-induced changes in the contractile properties of the BF [55]. According to previous findings [32,33,34], a longer Tc implies that fewer fast twitch fibers are activated during the TMG measurement protocol. Because changes in the activation pattern after a BF injury are detected as changes in the Tc and Td parameters from the TMG signal, the statistically significant differences in the TMG signal (Tc and Td) observed between injured and non-injured muscles are likely the result of an injury.

In the present investigation, there was no evidence of a statistically significant difference in the Dm when comparing the injured and non-injured muscles. This could have resulted from the variable time between the injury and TMG measurements (12 to 72 h) and the sensitivity of the signal to functional changes. For a more precise determination of the functional injury level, a stricter time window in which the measurements are performed (for example, 12–24 h after injury) would be required. This could be the topic of future investigations.

The ROC used to analyze the TMG diagnostic/classification showed the high discriminatory power of the procedure using the Tc percentage of the absolute differences for classification purposes. According to the data, the interval between 9.88% and 14.22% (Figure 7) of the absolute difference in the contraction time (injured/non-injured BF) is a critical cut-off range for values to distinguish injured from non-injured muscles. The test showed an excellent sensitivity of 98.1% and a specificity of 88.68% with a low (<1.9%) false negative ratio at the cut-off point of 12.50% of the Tc diff. When we changed the cut-off point (depending on priorities) to 9.89%, the sensitivity increased to 100% with a specificity of 86.8%. When analyzing female and male athletes within the injured and non-injured groups, no statistically significant differences were found in the ROC curves for the Tc classifier (difference −0.02, SE = 0.0223, z statistic = −0.897, and significance level *p* = 0.369) No Td or absolute percentage of differences was observed between the injured and non-injured BF muscles. Therefore, the Tc difference classifier is equally efficient for male and female subjects. The optimal cut-off point for females and males was also very similar (Figure 9, 12.50 and 13.57%). Additionally, no statistical differences were observed between the male and female Tc and Td of the BF in the non-injured group (Table 1). It is possible that the selection of athletes and similar long-term training adaptations converge the BF contractile properties of male and female athletes.

Finally, our study showed that the TMG signals of elite athletes of injured and non-injured BF are statistically different. Of all the TMG parameters analyzed (Tc, Td, Dm), Tc was the best classifier between injured and non-injured BF. Based on these facts, we reject the null hypothesis that there was no difference between the non-injured and injured BF muscles in elite athletes assessed by tensiomyography.

Considering possible future research, we can hypothesize that a higher number of TMG measurements in athletes before a BF injury during the same period of the competitive cycle would increase the predictive power of the measurements. One possible future use of TMG functional diagnostics, other than for the confirmation of BF injuries, could be in return to play strategies, especially as it allows for fast diagnostics (5–10 min to results) to be carried out continuously during the rehabilitation phase.

## 5. Limitations of the Study

This study took more than ten years to complete because the number of injured athletes was small, as was the number of athletes who qualified for the inclusion and exclusion criteria. A shorter study period and a significantly larger number of potential candidates for the inclusive/exclusive criteria would have contributed to a more accurate injury severity assessment. This study could be improved by performing the TMG measurements at any time within 20–24 h (limited interval) after the injury. In this way, the variability due to the time dependence of the muscle response after the injury would be avoided (in particular, the Dm parameter would be more useful). Ideally, we should have a reference value for the same muscle before the injury, which would require many more measurements.

## 6. Conclusions

The functional consequence of a biceps femoris injury is an increase in contraction time when applying twitch-type electrical stimulation. Based on the results of the present study investigation, the TMG Tc parameter can be used as an accurate binary classification method between injured and non-injured BF muscles (Tc diff female, AUC_f_ = 0.969, cut-off = 12.50%; Tc diff male, AUC_m_ = 0.989, cut-off = 13.57%). These results open the possibility of testing the same methodology/protocol for additional injury classification grades for BF injuries. It remains to be ascertained whether these findings can be extended to other muscles as well. Additionally, following a muscle injury, TMG diagnostics could be applied as a screening analysis/method or as a complementary tool to MRI/ultrasound or functional tests to optimize rehabilitation and post-rehabilitation procedures.

## Figures and Tables

**Figure 1 biology-11-00746-f001:**
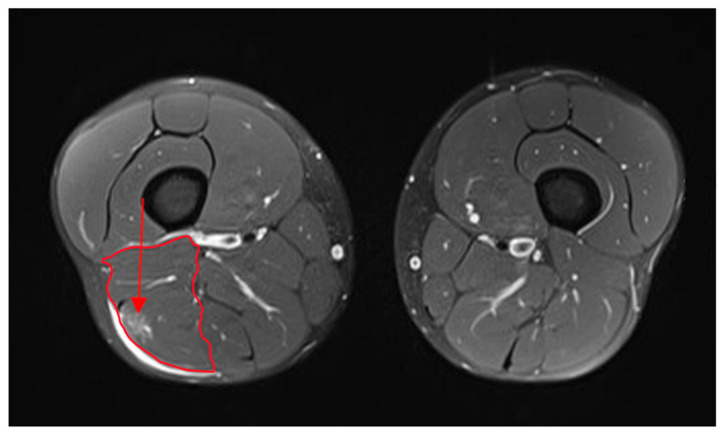
Example of magnetic resonance imaging, MRI of grade 1–2 BF injury. The pictures on the left show the left femur in cross-section with the injured biceps femoris (red circumscribed area), and the picture on the right shows the right femur in cross-section with the non-injured muscles of the same subject. Arrow indicates edema and location of the injury on the biceps femoris.

**Figure 2 biology-11-00746-f002:**
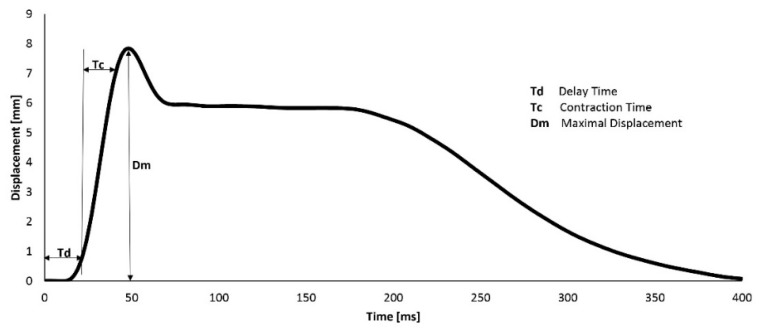
A typical example of a tensiomyography signal. Td (delay time): time from the beginning of the stimulus to a 10% increase in the amplitude; Tc (contraction time): time from 10 to 90% of the maximum amplitude; Dm, maximum amplitude of radial displacement. Reprinted from the *TMG Individual Report*, page 3. Copyright 2010 by TMG-BMC Ltd., Ljubljana, Slovenia.

**Figure 3 biology-11-00746-f003:**
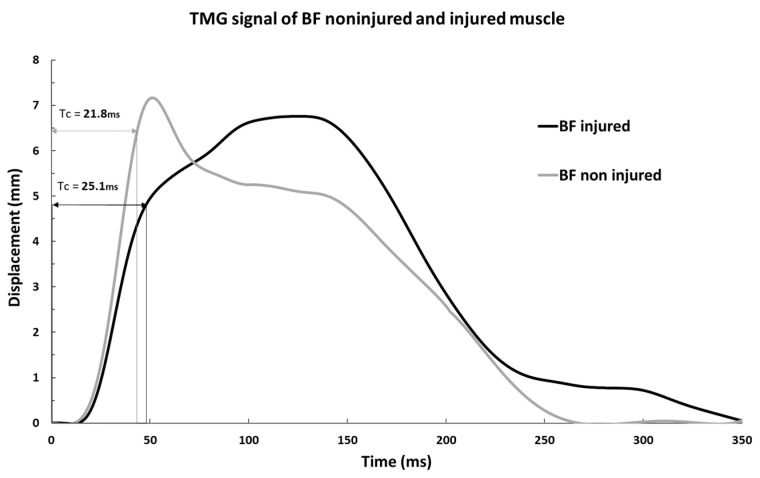
Characteristic TMG signal of injured and non-injured BF muscles from the same subject with Tc.

**Figure 4 biology-11-00746-f004:**
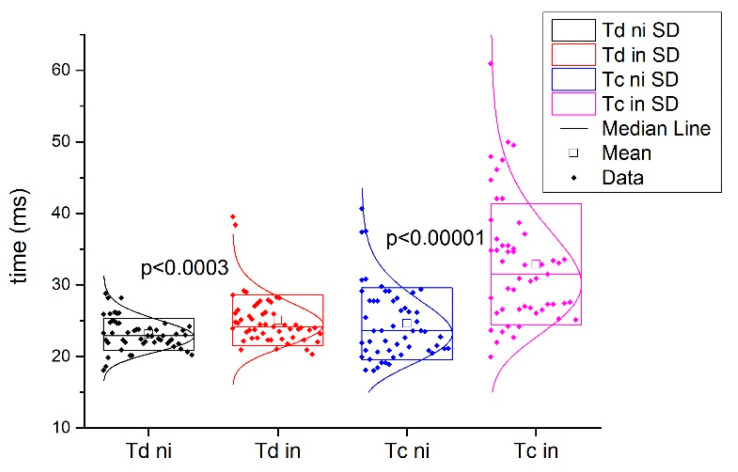
TMG parameters Td and Tc in the non-injured biceps femoris muscles of the female and male subjects and in the injured biceps femoris muscles of the female and male subjects; SD is the standard deviation.

**Figure 5 biology-11-00746-f005:**
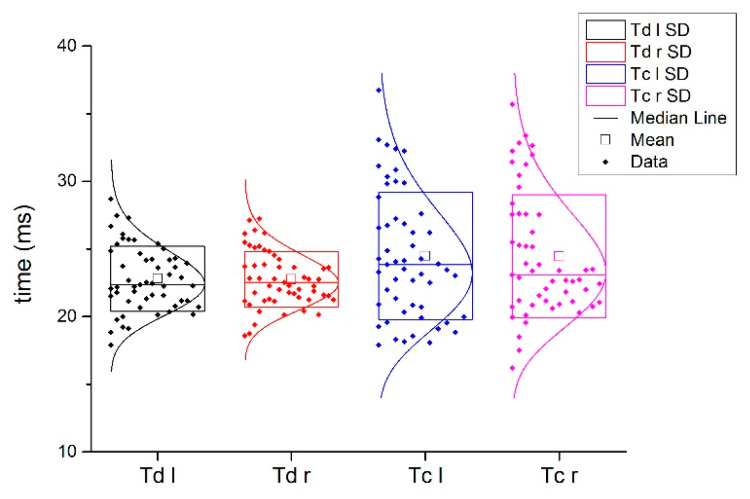
Comparison between Td l (delay time of left BF) and Td r (delay time of right BF) and between Tc l (contraction time of left BF) and Tc r (contraction time of right BF) in non-injured athletes. SD is the standard deviation.

**Figure 6 biology-11-00746-f006:**
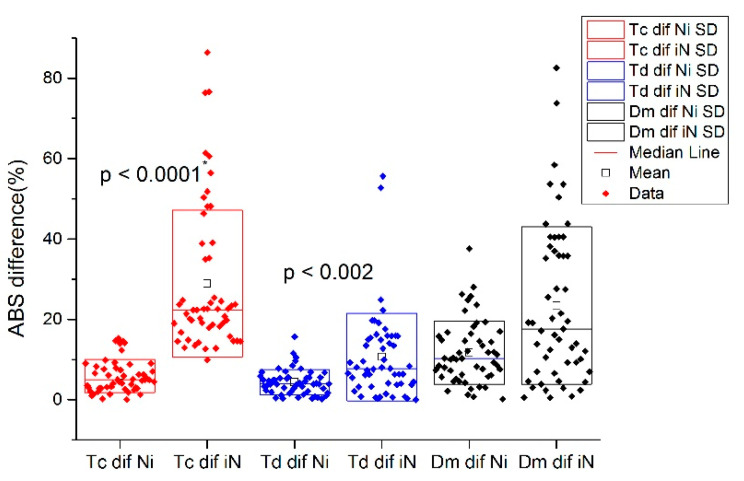
Comparison of absolute difference of percentages of BF Tc, Td, and Dm between non-injured and injured subjects.

**Figure 7 biology-11-00746-f007:**
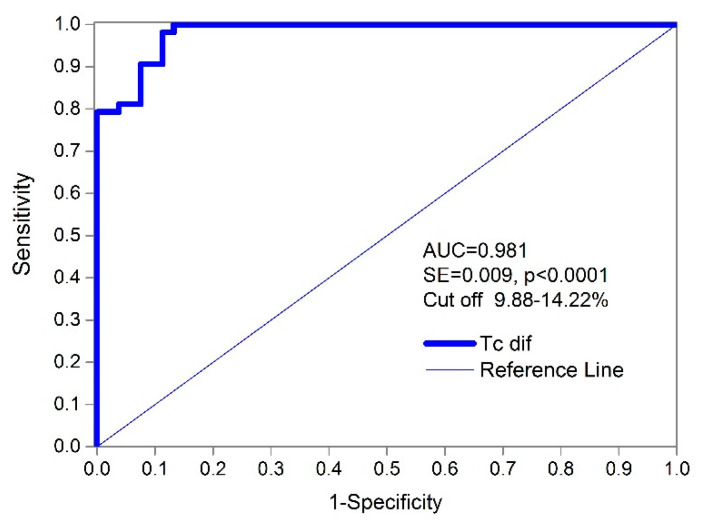
ROC curve for Tc diff in injured versus non-injured BF subjects.

**Figure 8 biology-11-00746-f008:**
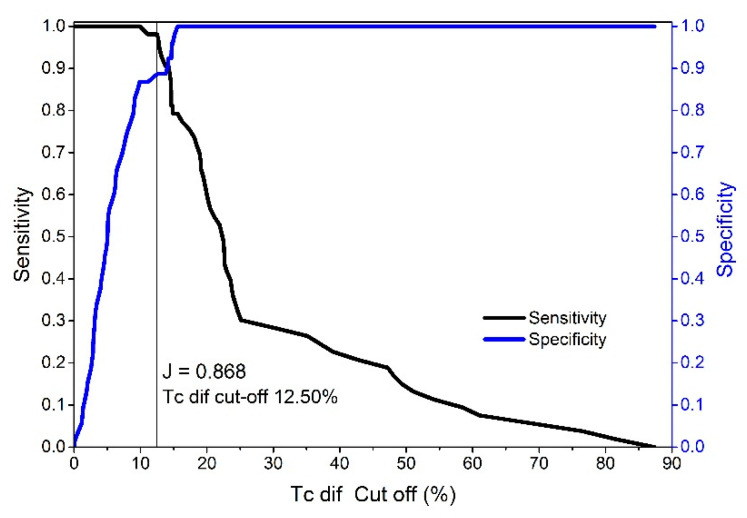
Cut-off classification criteria for injured and non-injured BF (the cut-off value of 12.50 is the best value for the Tc diff classifier using a combination maximum sum of diagnostically equally important sensitivity and specificity values).

**Figure 9 biology-11-00746-f009:**
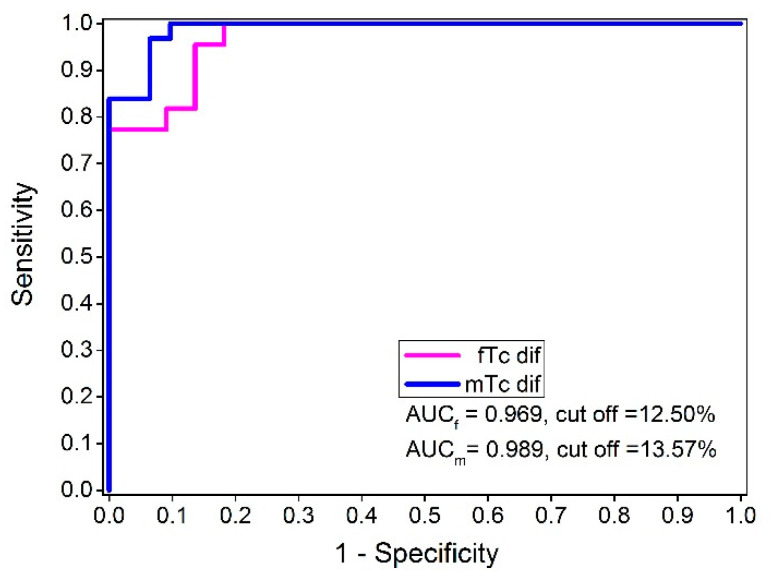
ROC curves of the female and male subgroups.

**Table 1 biology-11-00746-t001:** Descriptive statistical data and differences between variables (injured and non-injured BF and according to gender).

Female Non-Injured								
Td L (ms)	Td R (ms)	*p*-Value	Tc L (ms)	Tc R (ms)	*p*-Value	Dm L (mm)	Dm R (mm)	*p*-Value
23.1 ± 2.7	22.9 ± 2.1	0.466	23.9 ± 4.6	23.7 ± 3.7	0.736	4.1 ± 1.3	4.2 ± 1.2	0.327
Male Non-injured								
22.6 ± 2.2	22.7 ± 2.1	0.608	24.9 ± 4.8	25.0 ± 5.1	0.897	5.0 ± 2.0	5.1 ± 1.9	0.411
Injured subjects						Non-injured subjects		
Td ni (ms)	Td in (ms)	*p*-value				Td L (ms)	Td R (ms)	*p*-value
23.1 ± 2.3	25.0 ± 3.6	<0.001				22.8 ± 2.4	22.8 ± 2.1	0.819
Tc ni (ms)	Tc in (ms)	*p*-value				Tc L (ms)	Tc R (ms)	*p*-value
24.6 ± 5.1	32.9 ± 8.5	<0.001				24.5 ± 4.7	24.5 ± 4.5	0.882
Dm ni (mm)	Dm in (mm)	*p*-value				Dm L (mm)	Dm R (mm)	*p*-value
4.9 ± 1.8	5.0 ± 1.9	0.683				4.6 ± 1.8	4.8 ± 1.7	0.227
fTd ni (ms)	mTd ni (ms)	*p*-value				fTd L (ms)	mTd R (ms)	*p*-value
25.4 ± 3.8	24.8 ± 3.6	0.514				23.1 ± 2.9	23.1 ± 1.7	0.995
fTc ni (ms)	mTc ni (ms)	*p*-value				fTc L (ms)	mTc R (ms)	*p*-value
31.5 ± 7.6	33.9 ± 9.0	0.303				25.1 ± 5.9	24.2 ± 4.4	0.534
fDm ni (mm)	mDm ni (mm)	*p*-value				fDm L (mm)	mDm R (mm)	*p*-value
4.5 ± 2.0	5.4 ± 1.8	0.085				4.5 ± 1.9	5.2 ± 1.7	0.156
Non-injured group Female vs. Male								
fTd-L-ni	mTd-L-ni	*p*-value	fTc-L-ni	mTc-L-ni	*p*-value	fDm-L-ni	mDm-L-ni	*p*-value
23.1 ± 2.8	22.6 ± 2.2	0.464	23.9 ± 4.6	24.9 ± 4.8	0.406	4.1 ± 1.3	5.0 ± 2.0	0.049
fTd-R-ni	mTd-R-ni	data	fTc-R-ni	mTc-R-ni	data	fDm-R-ni	mDm-R-ni	data
22.9 ± 2.1	22.7 ± 2.1	0.746	23.7 ± 3.7	25.0 ± 5.1	0.298	4.2 ± 1.2	5.1 ± 1.9	0.035

Legend: Td L—delay time in left BF; Td R—delay time in right BF; Tc L—contraction time in left BF; Tc R—contraction time in right BF; Dm L—displacement in left BF; Dm R—displacement in right BF; Td ni—delay time in non-injured BF; Tc ni—contraction time in non-injured BF; Dm ni—displacement in non-injured BF; Td ni—delay time in non-injured BF in an injured subject; Td in—delay time in injured BF in an injured subject; Tc ni—contraction time in non-injured BF in an injured subject; Tc in—contraction time in injured BF in an injured subject; Dm ni—displacement in non-injured BF in an injured subject; Dm in—displacement in injured BF in an injured subject; fTd ni—delay time in non-injured BF in a female subject; mTd ni—delay time in non-injured BF in a male subject; fTc ni—contraction time in non-injured BF in a female subject; mTc ni—contraction time in non-injured BF in a male subject; fDm ni—displacement in a non-injured BF in a female subject; mDm ni—displacement in non-injured BF in a male subject; Td L—delay time in the left BF of non-injured subjects; Td R—delay time in the right BF of non-injured subjects; Tc L—contraction time in the left BF of non-injured subjects; Tc R—contraction time in the right BF of non-injured subjects; Dm L—displacement in the left BF of non-injured subjects; Dm R—displacement in the right BF of a non-injured subject; fTd L—delay time in the left BF of non-injured female subjects; mTd R—delay time in the right BF of non-injured male subjects; fTc L—contraction time in the left BF of non-injured female subjects; mTc R—contraction time in the right BF of non-injured male subjects; fDm L—displacement in the left BF of non-injured female subjects; mDm R—displacement in the right BF of non-injured male subjects; fTd-L-ni—delay time in the left BF of non-injured female subjects; mTd-L-ni—delay time in the left BF of non-injured male subjects; fTd-R-ni—delay time in the right BF of non-injured female subjects; mTd-R-ni—delay time in the right BF of non-injured male subjects; fTc-L-ni—contraction time in the left BF of non-injured female subjects; mTc-L-ni—contraction time in the left BF of non-injured male subjects; fTc-R-ni—contraction time in the right BF of non-injured female subjects; mTc-R-ni—contraction time in the right BF of non-injured male subjects; fDM-L-ni—displacement in left BF of non-injured female subjects mDm-L-ni—displacement in the left BF of non-injured male subjects; fDm-R-ni—displacement in the right BF of non-injured female subjects; mDm-R-ni—displacement in the right BF of non-injured male subjects.

**Table 2 biology-11-00746-t002:** Percentage results of absolute differences for variables according to injury criteria (injured vs. non-injured BF) and gender (male and female).

Absolute Differences between Injured vs. Non-Injured BF				
Variables	in (%)	ni (%)	χ^2^	*p*-Value
Td diff	7.80	3.93	14.21	<0.001
Tc diff	22.37	5.05	72.93	<0.001
Dm diff	17.63	10.27	8.58	<0.001
Absolute Differences Between Injured vs. Non-injured BF in Males				
mTd diff	6.35	3.93	9.12	<0.001
mTc diff	23.43	4.96	43.69	<0.001
mDm diff	14.02	10.33	1.81	0.179
Absolute Differences Between Injured vs. Non-injured BF in Females				
fTd diff	11.37	3.98	5.52	0.019
fTc diff	21.32	5.09	28.39	<0.001
fDm diff	19.91	10.18	8.34	<0.001
Injury Differences in Female vs. Male Subgroups				
	F in	M in	χ^2^	*p*-value
Td diff	11.37	6.35	1.02	0.312
Tc diff	21.32	23.43	2.76	0.097
Dm diff	19.91	14.02	1.42	0.233

**Table 3 biology-11-00746-t003:** ROC curve analysis results.

	Females—Injured vs. Non-Injured			Males—Injured vs. Non-Injured			Summarized F + M		
Variable	Dm	Tc	Td	Dm	Tc	Td	Dm	Tc	Td
AUC	0.754	0.969	0.707	0.599	0.989	0.723	0.665	0.981	0.712
St. Err.	0.076	0.021	0.086	0.076	0.009	0.068	0.055	0.009	0.052
95% CI	0.60–0.87	0.87–0.98	0.55–0.83	0.47–0.72	0.92–1.00	0.59–0.83	0.57–0.75	0.93–0.99	0.62–0.80
*p* value	0.001	<0.001	0.016	0.192	<0.001	<0.001	0.003	<0.001	<0.001
YI index	0.455	0.818	0.455	0.290	0.903	0.484	0.340	0.868	0.453
95% CI	0.23–0.59	0.64–0.91	0.23–0.64	0.13–0.39	0.74–0.97	0.26–0.65	0.17–0.45	0.75–0.93	0.28–0.59
Cut-Off	>25.81%	>9.25%	>7.76%	>17.00%	>9.87%	>5.91%	>17.0%	>9.87%	>5.91%
95% CI	19.14–25.81	6.14–15.36	1.51–15.71	6.98–37.64	7.76–14.63	5.52–10.53	10.10–28.04	8.82–14.18	5.52–11.59
Sensitivity	45.45	100.0	63.64	48.39	100.0	64.52	52.83	100.0	66.04
Specificity	100	81.82	81.82	80.65	90.32	83.87	81.13	86.79	79.25

Legend: Tables TMG parameters—Dm, Tc, and Td (absolute percentage of differences between injured and non-injured BF) were tested for the diagnostic accuracy of BF injury detection; ROC is a probability curve, and AUC represents the measure of separability; YI index is the J Youden index—J indicates the maximum performance of a given cut-off when the sensitivity and specificity reach the maximum score and when they are diagnostically equally important.

**Table 4 biology-11-00746-t004:** PRC curve analysis results.

	Females—Injured vs. Non-Injured			Males—Injured vs. Non-Injured			Summarized F + M		
Variable	Dm	Tc	Td	Dm	Tc	Td	Dm	Tc	Td
AUPRC	0.815	0.970	0.794	0.689	0.988	0.786	0.744	0.981	0.780
F1_max_	0.714	0.917	0.714	0.674	0.954	0.714	0.671	0.938	0.707
Association Criteria	>14.67	>9.25	>7.07	>0.19	>9.87	>5.91	>0.19	>9.87	>5.91
PPV	1.00	1.00	0.52	0.48	1.00	0.84	0.52	1.00	0.62
TPR	0.91	0.84	0.73	0.71	0.91	0.96	0.71	0.88	0.75

Legend: TMG parameters—Dm, Tc, and Td (absolute percentage of differences between injured and non-injured BF) were tested for the diagnostic accuracy of detecting BF injuries; PRC represents the precision–recall probability curve, and AUPRC represents the area under the precision–recall curve; F1_max_ indicates the measure of a test’s accuracy and is calculated from the precision (PPV—positive predictive value) and recall (sensitivity) of the test (TPR—true positive rate); Association Criteria is the criterion (measurement level) at which F1_max_ was reached.

## Data Availability

The data presented in this study are available on request from the corresponding author (S.Đ.). The data are not publicly available due to containing information that could compromise the privacy of research participants.

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
