# Peer review of "Tensiomyography Allows to Discriminate between Injured and Non-Injured Biceps Femoris Muscle"

_biology, 2022, doi:10.3390/biology11050746_

Round 1
Reviewer 1 Report
This is an interesting and well-designed study which aimed to test the hypothesis that the functional differences between injured and non-injured biceps femoris (BF) assessed by tensiomyography (TMG), a non-invasive mechanomyographic method, can be used for diagnostic and classification purposes. The authors argue that the rate of MRI-negative scans for clinically diagnosed hamstring injuries ranges from 14% to 45%, so hypothesize that TMG could serve to this purpose. The content of the article and the way of writing seem adequate and of quality. However, I have serious questions that must be answered by the authors, starting with the conflict of interest. After that, I will make some important notes on the statistics so that the authors can improve the information provided.
The following is very relevant and if the authors are not able to answer correctly, although the content is good, in my opinion it will deserve rejection by the journal. In the conflict of interest section the authors state: "the authors declare no conflict of interest". Is this really true? Since the authors chose an open-access journal, I have access to know who the authors are. In this regard, I have investigated and surprisingly, I have detected that at least 2 of the 5 authors (SD and SR) are related to the company that manufactures the TMG. SD is the co-owner and Head of R&D at TMG-BMC d.o.o. while SR acts as consultant at TMG-BMC d.o.o. (information obtained through Linkedin public profiles). These authors (SD and SR) together with another author (MP), are current members and founders of ISOT (International Society of Tensiomyography). Explain me why you stated no conflict of interest then because I obviously see the opposite.
Regarding the content of the article, major considerations:
Material and methods. Please, the Research ethics committee reference number should be stated. If an ethics committee passed, it must be present.
Statistics. Receiver operating characteristic (ROC) curves show how well a risk prediction model discriminates between subjects with and without a condition while the area under the curve (AUC) is the measure of the ability of a classifier to distinguish between classes and is used as a summary of the ROC curve. The authors used both to measure the accuracy of the classification and to identify cut-off limits using time of contraction (Tc) differences. While the employed statistics are correct, class imbalance is a common problem in the binary predictive modeling space. I strongly suggest authors add the Precision-Recall Curve (PRC). PRC summarize the trade-off between the true positive rate and the positive predictive value for a predictive model using different probability thresholds. When the negative class is more prevalent and there is low value in true-negative predictions, Precision-Recall curve is preferred over ROC, so the inclusion of PRC is more informative than ROC in imbalanced data. This journal has no restrictions on the maximum length of research manuscripts, so authors can include it in the main text or add it as supplemental material. Be it one way or another, this should be calculated since they are complementary methods.
Minor considerations:
Introduction. Line 56, change "electromyography" to "electromyographic". Lines 60-68, please be pragmatic and non-redundant, when citing the reliability studies is enough to refer to the reviews (since reviews include de original cited articles). In addition, please expand the introduction on these lines a bit by talking about the main differences between the types of MMG sensors (contact, laser...) and if these types of methods are comparable between them (they measure the same or similar)? This information is constructive for novel readers on the topic. Line 69, change "injuries" to "(e.g., injuries)" since a functional and structural muscle disorder is not only an injury, but an illness too.
Material and methods. Why did you only measured three TMG parameters if there are more? This should be justified in the text, for example, arguing based on the reliability reviews, that the Tr and Ts parameters are not reproducible. Why the velocity of radial displacement (Vrd) also bad called velocity of contraction (Vc) was not added since integrates several TMG responses?
Line 84, "(...) were placed symmetrically 2 cm from the sensor". Symmetrically? The sensor is not positioned symmetrically with respect to the electrodes in Figure 1, how can I as a reviewer believe that the measurements were correctly done? It has been tested in the literature what happens if this is not symmetrically tested? I should trust that the measurements were well performed, but understand my concerns as reviewer. Please, do not include this figure in the article. In place of this figure, for example, authors may include an MRI or ultrasound image of some subject's injured vs. uninjured leg. It should be an MRI, since is arguably the gold standard imaging technique in muscle injury detection (10.1259/bjr/84622172).
Statistics. Did the authors calculated the a priori sample size? This is quite important in this type of study where you are trying to argue that A (TMG) is better than B (MRI) in detecting something, when B is the gold standard.
Author Response
REWIEV 1
Open Review
(x) I would not like to sign my review report
( ) I would like to sign my review report
English language and style
( ) Extensive editing of English language and style required
( ) Moderate English changes required
(x) English language and style are fine/minor spell check required
( ) I don't feel qualified to judge about the English language and style
|
Yes |
Can be improved |
Must be improved |
Not applicable |
|
|
Does the introduction provide sufficient background and include all relevant references? |
( ) |
(x) |
( ) |
( ) |
|
Is the research design appropriate? |
(x) |
( ) |
( ) |
( ) |
|
Are the methods adequately described? |
( ) |
(x) |
( ) |
( ) |
|
Are the results clearly presented? |
(x) |
( ) |
( ) |
( ) |
|
Are the conclusions supported by the results? |
(x) |
( ) |
( ) |
( ) |
Comments and Suggestions for Authors
This is an interesting and well-designed study which aimed to test the hypothesis that the functional differences between injured and non-injured biceps femoris (BF) assessed by tensiomyography (TMG), a non-invasive mechanomyographic method, can be used for diagnostic and classification purposes. The authors argue that the rate of MRI-negative scans for clinically diagnosed hamstring injuries ranges from 14% to 45%, so hypothesize that TMG could serve to this purpose. The content of the article and the way of writing seem adequate and of quality. However, I have serious questions that must be answered by the authors, starting with the conflict of interest. After that, I will make some important notes on the statistics so that the authors can improve the information provided.
The following is very relevant and if the authors are not able to answer correctly, although the content is good, in my opinion it will deserve rejection by the journal. In the conflict of interest section the authors state: "the authors declare no conflict of interest". Is this really true? Since the authors chose an open-access journal, I have access to know who the authors are. In this regard, I have investigated and surprisingly, I have detected that at least 2 of the 5 authors (SD and SR) are related to the company that manufactures the TMG. SD is the co-owner and Head of R&D at TMG-BMC d.o.o. while SR acts as consultant at TMG-BMC d.o.o. (information obtained through Linkedin public profiles). These authors (SD and SR) together with another author (MP), are current members and founders of ISOT (International Society of Tensiomyography). Explain me why you stated no conflict of interest then because I obviously see the opposite.
Regarding the content of the article, major considerations:
Thank you for this comment.
Thanks to the reviewer for noticing and commenting on the non-existence, i.e. omission in terms of a statement of conflict of interest.
Authors explanation:
Generally, the authors did not intend to hide anything but duly reported their affiliations;
The authors belong to the category of cited researchers with a large number of already published papers (PZ, NM, S.Dj., MD) in the field of TMG in highly indexed journals (J Strength Cond Res, Sensors, J Biomechanics, Cell Mol Biol Lett, etc.), work on the university, are engaged in education in medical and sports fields, so publishing one paper is no motive for compromising their professional integrity. It was a just a matter of technical and administrative omission in the sense of one sentence (a statement of conflict of interest);
Finally, the authors fully agree to add a statement in terms of conflict of interest to protect the journal's integrity and the MDPI company.
Material and methods. Please, the Research ethics committee reference number should be stated. If an ethics committee passed, it must be present.
Thank you for this comment.
In the page 13, at lanes 423 and 424 were already two reference numbers from the ethics committee from Slovenia (Commission of the Republic of Slovenia for Medical Ethics) and the University of Belgrade, Serbia, as well as:
Originally submitted manuscript version:
Institutional Review Board Statement: The study was conducted according to the guidelines of the Declaration of Helsinki and approved by the Commission of the Republic of Slovenia for Medical Ethics, approval study number - No. 125/03/14, and by the Institutional Ethical Board of Faculty of Sport and Physical Education, University of Belgrade Serbia, approval study number - 484-2.
Action:
At the beginning of Methodes, as a second paragraf was added:
The study was approved by the Commission of the Republic of Slovenia for Medical Ethics, approval study number - No. 125/03/14, and by the Institutional Ethical Board of Faculty of Sport and Physical Education, University of Belgrade Serbia, approval study number - 484-2.
Comment:
Statistics. Receiver operating characteristic (ROC) curves show how well a risk prediction model discriminates between subjects with and without a condition while the area under the curve (AUC) is the measure of the ability of a classifier to distinguish between classes and is used as a summary of the ROC curve. The authors used both to measure the accuracy of the classification and to identify cut-off limits using time of contraction (Tc) differences. While the employed statistics are correct, class imbalance is a common problem in the binary predictive modeling space. I strongly suggest authors add the Precision-Recall Curve (PRC). PRC summarize the trade-off between the true positive rate and the positive predictive value for a predictive model using different probability thresholds. When the negative class is more prevalent and there is low value in true-negative predictions, Precision-Recall curve is preferred over ROC, so the inclusion of PRC is more informative than ROC in imbalanced data. This journal has no restrictions on the maximum length of research manuscripts, so authors can include it in the main text or add it as supplemental material. Be it one way or another, this should be calculated since they are complementary methods.
Action:
We do include PRC analysis as a ROC complementary method in 2.5. Statistical analysis showed all PRD statistical data with explanation in new Table 3 and one new paragraph placed at the and of the 3. Results chapter.
Minor considerations:
Introduction. Line 56, change "electromyography" to "electromyographic". Lines 60-68, please be pragmatic and non-redundant, when citing the reliability studies is enough to refer to the reviews (since reviews include de original cited articles). In addition, please expand the introduction on these lines a bit by talking about the main differences between the types of MMG sensors (contact, laser...) and if these types of methods are comparable between them (they measure the same or similar)? This information is constructive for novel readers on the topic. Line 69, change "injuries" to "(e.g., injuries)" since a functional and structural muscle disorder is not only an injury, but an illness too.
Action
It was added in introductions 66-72, by the suggestion in the reviewer's suggestion.
Line 69 was changed (now 81)
Comment:
Line 56, change "electromyography" to "electromyographic".
Action:
The change was done by the suggestion in the reviewer's comment.
Comment:
Lines 60-68, please be pragmatic and non-redundant, when citing the reliability studies is enough to refer to the reviews (since reviews include de original cited articles).
Action:
The change was done following the reviewer's comment. We excluded two references from the cited list (26 and 27) and (28 and 29).
Comment:
Line 69, change "injuries" to "(e.g., injuries)" since a functional and structural muscle disorder is not only an injury, but an illness too.
Action:
The change was done following the reviewer's comment.
Material and methods. Why did you only measured three TMG parameters if there are more? This should be justified in the text, for example, arguing based on the reliability reviews, that the Tr and Ts parameters are not reproducible. Why the velocity of radial displacement (Vrd) also bad called velocity of contraction (Vc) was not added since integrates several TMG responses?
Line 84, "(...) were placed symmetrically 2 cm from the sensor". Symmetrically? The sensor is not positioned symmetrically with respect to the electrodes in Figure 1, how can I as a reviewer believe that the measurements were correctly done? It has been tested in the literature what happens if this is not symmetrically tested? I should trust that the measurements were well performed, but understand my concerns as reviewer. Please, do not include this figure in the article. In place of this figure, for example, authors may include an MRI or ultrasound image of some subject's injured vs. uninjured leg. It should be an MRI, since is arguably the gold standard imaging technique in muscle injury detection (10.1259/bjr/84622172).
Comment:
Why did you only measured three TMG parameters if there are more? This should be justified in the text, for example, arguing based on the reliability reviews, that the Tr and Ts parameters are not reproducible. Why the velocity of radial displacement (Vrd) also bad called velocity of contraction (Vc) was not added since integrates several TMG responses?
Action:
The change was done in accordance with the reviewer's comment. We included folowing explanation: We used only highly reproducible parameters based on previous reliability reviews [28,29].
Comment:
Line 84, "(...) were placed symmetrically 2 cm from the sensor". Symmetrically? The sensor is not positioned symmetrically with respect to the electrodes in Figure 1, how can I as a reviewer believe that the measurements were correctly done? It has been tested in the literature what happens if this is not symmetrically tested? I should trust that the measurements were well performed, but understand my concerns as reviewer. Please, do not include this figure in the article. In place of this figure, for example, authors may include an MRI or ultrasound image of some subject's injured vs. uninjured leg. It should be an MRI, since is arguably the gold standard imaging technique in muscle injury detection (10.1259/bjr/84622172).
Action:
The change was partially done following the reviewer's comment.
Authors explanation: After a discussion among the authors, it was concluded that the best solution is to include MRI of BF injury image inthe text (Figure 1).
Statistics. Did the authors calculated the a priori sample size? This is quite important in this type of study where you are trying to argue that A (TMG) is better than B (MRI) in detecting something, when B is the gold standard.
Action:
All necessary new text was added, considring power and sample size calculation (2.5 Statistics chapter) following the reviewer's comment.
Submission Date
26 January 2022
Date of this review
02 Feb 2022 14:57:14

Reviewer 2 Report
This is an interesting study and is overall well written, congratulations, but requires improvements and a careful review later, assuming the instructions for authors of the journal. Below contributions with line indication.
4 - Affiliation 4 should be changed with 7, respecting the order of appearance.
20 & 28 - MRI in full suggested.
48-75 - Please consider developing the introduction section.
72 - Please indicate the aim or aims of the study.
79 – Please indicate the institutional ethical code (it is in lines 421-424).
105 – Please describe the inclusion and exclusion criteria and how was the sample selected (local club? Convenience?).
105 - More information is necessary about the subjects (height, weight, BMI, years of practice, training routines…).
121 - More information is necessary regarding data collection. Time of day and possible influence in the results? The local, temperature, humidity, and other conditions. Who followed the procedures for data collection, their roles and experience?
156 - Please provide statistical power information.
185 & 195 - Please verify if the journal template assumes a space between text.
224 - A legend is suggested for table 1.
224 & 251 - Please consider differentiating the significance of the values aiming readers comprehension.
262 - Please consider a better-quality image (also fig 7, 8, 9)
265 - ROC suggested in full.
266 - Please review the legend content.
271-299 - Please consider text reorganization. The paragraph is very extensive.
306 - Please consider directing the discussion further towards faster and better reading and interpretation by manuscript readers.
309 - 326 - The citation references are different (space in commas). Please review in these lines and throughout all the manuscript.
396 - Please consider describing study limitations and suggestions for future research.
397 - Conclusions should be more summarized and objective. Practical applications indication may be considered.
432 - Please review all the references format according to the journal instructions for authors. Normally “;” between names and “,” after surnames. Also points between initials should be placed. Another example, 520 ref year is not in bold.
Author Response
REWIEV 2
Open Review
(x) I would not like to sign my review report
( ) I would like to sign my review report
English language and style
( ) Extensive editing of English language and style required
(x) Moderate English changes required
( ) English language and style are fine/minor spell check required
( ) I don't feel qualified to judge about the English language and style
|
Yes |
Can be improved |
Must be improved |
Not applicable |
|
|
Does the introduction provide sufficient background and include all relevant references? |
( ) |
( ) |
(x) |
( ) |
|
Is the research design appropriate? |
( ) |
(x) |
( ) |
( ) |
|
Are the methods adequately described? |
( ) |
( ) |
(x) |
( ) |
|
Are the results clearly presented? |
( ) |
(x) |
( ) |
( ) |
|
Are the conclusions supported by the results? |
( ) |
(x) |
( ) |
( ) |
Comments and Suggestions for Authors
This is an interesting study and is overall well written, congratulations, but requires improvements and a careful review later, assuming the instructions for authors of the journal. Below contributions with line indication.
Comment:
4 - Affiliation 4 should be changed with 7, respecting the order of appearance.
Action:
All change was done following the reviewer's comment.
Comment:
20 & 28 - MRI in full suggested.
Action:
The all change was done following the reviewer's comment.
Comment:
48-75 - Please consider developing the introduction section.
Action:
We added two sentence as a better introduction for the mentioned section, according th the reviewer's comment. Now, the first paragraph in the Introduction chapter starts as follows:
Top athletes have daily and even multiple daily training sessions that are realized in different intensity zones. In addition to training, they also have regular competitive efforts during the annual cycles. Because of that, elite sports performance is associated with a high musculoskeletal soft tissue injury risk. .......
Comment:
72 - Please indicate the aim or aims of the study.
Action:
The all change was done in accordance with the reviewer's comment. We added new sentence with aim of the study at the and of the Introduction chapter as follow:
Based on the previous comments, it can be concluded that the primary aim of this study is to test the null hypothesis of no difference in injured and non-injured BF con-traction properties in elite athletes assessed by tensiomyography. The secondary aim was to investigate the highest sensitivity and specificity of TMG classifier to separate between injured and non-injured BF muscles by applying optimal cut-off values.
Comment:
79 – Please indicate the institutional ethical code (it is in lines 421-424).
Action:
All changes was done following the reviewer's comment. We added a new sentence with the ethical codes and the information considering the Ethical Institution Commission approvement for study. We added new sentence as a second paragraph at the Materials and Methods chapter as follows:
The study was approved by the Commission of the Republic of Slovenia for Medical Ethics, approval study number - No. 125/03/14, and by the Institutional Ethical Board of Faculty of Sport and Physical Education, University of Belgrade Serbia, approval study number - 484-2.
Comment:
105 – Please describe the inclusion and exclusion criteria and how was the sample selected (local club? Convenience?).
Action:
The all changes was done in accordance with the reviewer's comment. We added new sentence with the with better explanation for including and excluding criteria for the study. We added new sentence as a last paragraph at a Subjects chapter as follow:
The criteria for including subjects in the study were as follows: subjects were adult athletes of national and international rank who played in the first team (football players) or competed in the first team (track-and-field athletes) of their clubs. The criterion for excluding subjects from the study was a history of a hamstring injury for at least two years and a history of chronic low back pain.
Comment:
105 - More information is necessary about the subjects (height, weight, BMI, years of practice, training routines…).
Action:
All changes was done following the reviewer's comment. We added two new sentences with all basic anthropometric and training characteristics data for the study. We added one sentence in first and at third paragraph at a Subjects chapter as follows:
- (22 females and 31 males, Age - 24.2 ± 4.6 years, BH - 180.4 ± 7.6 cm; BM - 70.9 ± 8.4 kg; BMI – 21.72 ± 1.28 kg/m2; training experience – 14.0 ± 5.2 years; number of training 7.8 ± 1.2 training/week)
- (22 females and 31 males, Age 23.4 ± 4.3 years, BH - 181.9 ± 6.9 cm; BM - 70.9 ± 7.0 kg; BMI – 21.39 ± 0.83 kg/m2; training experience – 12.6 ± 4.3 years; number of training 7.4 ± 1.3 training/week)
Comment:
121 - More information is necessary regarding data collection. Time of day and possible influence in the results? The local, temperature, humidity, and other conditions. Who followed the procedures for data collection, their roles and experience?
Action:
Was added (105-108)
The subjects had no organized physical activity for at least 60 minutes before the TMG measurements. TMG measurements were always taken between 10 am and 4 pm in a temperature-controlled room. The temperature during the measurements was between 21 and 25°C. We do not have humidity data.
The data collection procedure was performed/controlled by PZ, SD, SR, MD, all with at least 15-3 years of experience ( Kinesiology, Biomechanics of skeletal muscle, Sports medicine) and published scientific articles.
Comment:
156 - Please provide statistical power information.
Action:
All necessary new text was added, considring statistical power and sample size calculation (2.5 Statistics chapter) following the reviewer's comment.
Comment:
185 & 195 - Please verify if the journal template assumes a space between text.
Action:
The mentioned part of the manuscript is checked. We assume that due to the writing of formulas, the space text has been changed. We suppose that this is following the technical conditions for writing formula forms.
Comment:
224 - A legend is suggested for table 1.
Action:
All changes was done following the reviewer's comment. The Legend is added below the Tabel 1.
Comment:
224 & 251 - Please consider differentiating the significance of the values aiming readers comprehension.
Action:
The entire text has been reviewed in detail and the numerical values of statistical significance have been harmonized following the reviewer's comment.
Comment:
262 - Please consider a better-quality image (also fig 7, 8, 9)
Action:
Figures 7, 8 and 9 were transformed into figures 6, 7 and 8, and it was changed with high qualities.
Comment:
265 - ROC suggested in full.
Action:
The entire text has been reviewed in detail and the ROC curve data have been corrected. Also, extra analyze - Precision-Recall Curve (PRC) data was added to the text (Table 4).
Comment:
266 - Please review the legend content.
Action:
The entire text has been reviewed in detail and the legend contents in all Tables have been harmonized following the reviewer's comment.
Comment:
271-299 - Please consider text reorganization. The paragraph is very extensive.
Action:
We accepted in total reviewer's suggestion. We created two paragraphs and made proper corrections to the text.
Comment:
306 - Please consider directing the discussion further towards faster and better reading and interpretation by manuscript readers.
Action:
We fully accepted the reviewer's suggestion, and the entire text has been reviewed in detail again with all made proper corrections.
Comment:
309 - 326 - The citation references are different (space in commas). Please review in these lines and throughout all the manuscript.
Action:
We fully accept the reviewer's suggestion. We made proper corrections in the text, considering references citations in the text.
Comment:
396 - Please consider describing study limitations and suggestions for future research.
Action 1:
We accepted in total reviewer's suggestion. We added a brand new paragraph as a last one at the Discussion chapter:.
Considering possible future research, we can hypothesize that a higher number of measurements in athletes who had a TMG measurement of BF before injury during the same period of the competitive cycle would increase the predictive power of the measurements. One possible future use of TMG functional diagnostics, other than confirmation of BF injuries, could be in the application in return to play strategy, especially as it allows fast diagnostics (5-10 min to results) to be carried out continuously into the rehabilitation phase.
Action 2:
We accepted in fully reviewers sugestion. We added brand new paragraph at the Limitations chapter as follow:
The study took more than ten years because the resources of injured athletes were limited, as was the number of athletes who qualified for the inclusion and exclusion criteria. A shorter study period and a significantly larger number of potential candidates for the inclusive/exclusive criteria would have contributed to a more accurate assessment of injury severity. An improvement would be to perform TMG measurements at any time 20-24h (limited interval) after injury. In this way, the variability due to the time dependence of the muscle response after injury would be avoided (in particular, the Dm parameter would be more useful). Ideally, we should have a reference value of the same muscle before the injury, requiring many more measurements.
Comment:
397 - Conclusions should be more summarized and objective. Practical applications indication may be considered.
Action:
We accepted reviewers' suggestions. We made the Conclusion chapter more understandable and shorter, emphasizing practical application.
Comment:
432 - Please review all the references format according to the journal instructions for authors. Normally ";” between names and “,” after surnames. Also points between initials should be placed. Another example, 520 ref year is not in bold.
Action:
We accepted reviewers' suggestions. We made the full References chapter check and made all necessary corrections.
Submission Date
26 January 2022
Date of this review
04 Feb 2022 15:53:01

Reviewer 3 Report
Overall, this is written quite well. I do suggest that this should be submitted as a brief report rather than an article. I have only a few suggestions:
Put the subject section first in the methods.
Authors should disclose a power analysis to justify their sample size.
P values should not be noted at "0.000" but rather "<0.001"
In the conclusions, I suggest authors include the limitations/future research in one single paragraph, then give another paragraph with the main findings.
Author Response
REVIEW 3
Open Review
(x) I would not like to sign my review report
( ) I would like to sign my review report
English language and style
( ) Extensive editing of English language and style required
( ) Moderate English changes required
(x) English language and style are fine/minor spell check required
( ) I don't feel qualified to judge about the English language and style
|
Yes |
Can be improved |
Must be improved |
Not applicable |
|
|
Does the introduction provide sufficient background and include all relevant references? |
(x) |
( ) |
( ) |
( ) |
|
Is the research design appropriate? |
(x) |
( ) |
( ) |
( ) |
|
Are the methods adequately described? |
(x) |
( ) |
( ) |
( ) |
|
Are the results clearly presented? |
(x) |
( ) |
( ) |
( ) |
|
Are the conclusions supported by the results? |
(x) |
( ) |
( ) |
( ) |
Comments and Suggestions for Authors
Overall, this is written quite well. I do suggest that this should be submitted as a brief report rather than an article. I have only a few suggestions:
Comment:
Put the subject section first in the methods.
Action:
We accepted reviewers' suggestions. We moved the Subjects section as a first in the Methods chapter and created a new subtitle 2.2. Measurement Procedures
Comment:
Authors should disclose a power analysis to justify their sample size.
Action:
All correction considering power analysis was done. We added the following text in chapter 2.5. Statistics:
The sample size and power analysis were calculated by G*Power 3.1.9.4 statistical software (Franc Faul, University of Kiel, Germany, ©1992-2019). Results showed that for two groups and total sample size at 106 effect size d for 0.2 - small, 05 – medium, and 0.8 – large power is at 0.175, 0.722 and 0.983, respectively.
Comment:
P values should not be noted at "0.000" but rather "<0.001"
Action:
We accepted reviewers' suggestions. We changed all 0.000 marks in <0.001 (which was mainly in the Tables).
Comment:
In the conclusions, I suggest authors include the limitations/future research in one single paragraph, then give another paragraph with the main findings.
Action:
We included a brand new chapter: 5. Limitations of the study after 4. Discussion and, and before chapter 6. Conclusions.
Submission Date
26 January 2022
Date of this review
28 Feb 2022 15:35:48

Reviewer 4 Report
Tensiomyography allows to assess injuries of the biceps femoris and discriminate between injured and non-injured muscle
This is an interesting study showing that evoked twitches have a longer Tc with TMG in injured compared to uninjured BF femoris, at least when this is measured within the 72 hours after sustaining the injury. Tc has high discriminatory power. The data analysis seems fine. But I do question the many t-tests that are use. Moreover, the presentation can be improved. There are hardly any important differences between males and females, but reporting the data of both in detail doesn’t enhance the readability.
The impact of the work could be enhanced if the authors succeed in convincing the reader (reviewer) that indeed measuring Tc is of added value, to me it seems that this all remains to be seen because simply asking the athlete soon within 3 days after sustaining an hamstring injury: ‘which leg hurts?’ also has a very high discriminatory power….
This is a clear abstract, the first question that I would have: how soon after the injury occurred were the athletes measured? Maybe this information could be added in the abstract?
In the introduction I do miss a clear incentive for testing H0. I read about hamstring injuries and about TMG and the link seems to be given in this single sentence: Functional and structural muscle disorders (injuries) result in functional changes of the injured muscle
But I still would ask: what is the problem and how exactly could establishing a difference in contractile properties between injured and uninjured muscle help? (I can think of some reasons, but these are not provided in the introduction.
Probably some kind of file number should be added to the statement of ethical approval being provided
With the electrical stimulation how was current strength set and/or is the same strength used in for both legs and /or in all participants? Please add this information
I see this is provided in line 127. If the current increase is stopped when pain is evoked, doesn’t this introduce the risk that currents (and thereby muscle force and probably Dm are on average lower for the injured compared to the healthy leg? This is an important issue I would think.
Were the examiners during testing aware of whether the participant was injured or not and/or in which leg the injury occurred? Please add this information, but after reading the next paragraph I think that blind assessment did not occur….turns out it did (lines 118-120) , which is a strength of the design, although I have some doubts whether this truly possible with a serious injury (as soon as the athlete enters the room you can see which leg is hurt, can’t you?)
Line 100-113 Do I understand correctly that the TMG was performed within a few days after the athletes sustained the injury? This makes my question about the introduction even more relevant, because if we can measure differences between the legs which doesn’t seem unlikely so soon after an injury) what would be next/how would this help recovery or what? If we would want to know which leg is injured, we could just ask the athlete. So you really should better explain, what maybe is obvious to you but not for the average reader, what the benefit of doing these measurements would be.
The acronyms line 141-153 are too many for this reviewer.
fTc diff ni = percentage of the absolute difference between non-injured left and right BF Tc in the female subgroup? I will have to study line 179 to understand this…
Are all these necessary? I would stick to the main question why would one introduce left and right? Injured versus uninjured is of interest, isn’t it? Do you expect differences (for the differences between injured uninjured) between males and females?
I hope that you are not going to test for all these potential differences….this inflated your statistical power. In table one I see that many tests are performed. Did you correct for the number of t-tests performed? With so many tests and variables, you will always find a difference somewhere by chance.
Lines 179-191.., isn’t this just expressing the differences between injured and uninjured as a percentage of the mean of the maximal values of both legs ?
Line 192: Okay now I do understand that you need left and right for a negative control (still the acronyms scare me off, these do not increase the readability of your work)
Results
I do not think you can report the results with two significant digits (32.88 ms) this suggest an accuracy that certainly can’t be present can it? 32.9 will do for the mean values
Table 1 is quite a puzzle I see two time ni on the left side for Td Tc Dm etc. shouldn’t that be ni versus in ? This is what a meant by indicating not liking the acronyms it seems that even the authors themselves got confused….
Figure 3. I am really curious about the explanation for the differences in Displacement traces around 50 ms between injured and ni. Interesting! The hump in the in-trace at 250 ms, suggests some kind of additional (reflex, or persistent inward current phenomenon) activity in the injured BF.
Figure 4 Good figure but in the caption it reads of the female and ae subjects, so I was looking for separate female and male data, but you probably mean: of all subjects (male and female together)
I think the readability of the result section would increase by only reporting the parameters of the AUC analysis for Tc in the text (deleting lines 281-292)
Figure 6 and table 3 clearly show that only Tc has discriminatory potential
Discussion
Lines 307-340 Part of this discussion should be included in the introduction to make the objective clear.
This all makes sense to me, but I still am left with my main question. So great now we can objectively establish in the early stage after sustaining an injury, that there is an injury, but it is still unclear to me how this would help the rehabilitation process?
Lines 341-356 Okay fine I fully agree, but line 357: In this way, repeated assessments of
large groups are possible, allowing the collection of clinically relevant longitudinal data.
In answer to my question about Figure 3; what causes the differences? I read lines 360-372, which is interesting. Could swelling due to oedema also play a role? If that is the case the differences in Tc between in and ni may indeed disappear relatively early in the rehabilitation period (and wouldn’t have little added value in the prognostics or treatment)
Author Response
REWIEV 4
Open Review
(x) I would not like to sign my review report
( ) I would like to sign my review report
English language and style
( ) Extensive editing of English language and style required
( ) Moderate English changes required
(x) English language and style are fine/minor spell check required
( ) I don't feel qualified to judge about the English language and style
|
Yes |
Can be improved |
Must be improved |
Not applicable |
|
|
Does the introduction provide sufficient background and include all relevant references? |
( ) |
( ) |
(x) |
( ) |
|
Is the research design appropriate? |
(x) |
( ) |
( ) |
( ) |
|
Are the methods adequately described? |
(x) |
( ) |
( ) |
( ) |
|
Are the results clearly presented? |
( ) |
(x) |
( ) |
( ) |
|
Are the conclusions supported by the results? |
( ) |
( ) |
(x) |
( ) |
Comments and Suggestions for Authors
Tensiomyography allows to assess injuries of the biceps femoris and discriminate between injured and non-injured muscle
This is an interesting study showing that evoked twitches have a longer Tc with TMG in injured compared to uninjured BF femoris, at least when this is measured within the 72 hours after sustaining the injury. Tc has high discriminatory power. The data analysis seems fine. But I do question the many t-tests that are use. Moreover, the presentation can be improved. There are hardly any important differences between males and females, but reporting the data of both in detail doesn’t enhance the readability.
Comment:
The impact of the work could be enhanced if the authors succeed in convincing the reader (reviewer) that indeed measuring Tc is of added value, to me it seems that this all remains to be seen because simply asking the athlete soon within 3 days after sustaining an hamstring injury: ‘which leg hurts?’ also has a very high discriminatory power….
General Comment and authors Response:
We are grateful for reviewer comments and suggestions. But, we have a standing point that the difference in Tc is quantitative, while pain is a qualitative difference. Not all pain (in the hamstrings region) indicates a BF muscle injury. The pain sensation is a subjective phenomenon and does not allow a precise location/type of the injury per se. It may be an adjacent muscle (semitendinosus, semimembranosus) ligament, fascia, etc.
Comment:
This is a clear abstract, the first question that I would have: how soon after the injury occurred were the athletes measured? Maybe this information could be added in the abstract?
General Comment and authors Response:
Thank you for the reviewer comment. The suggestion and author's comment about interval were mentioned in the Disscusion chapter, and in the Limitation of the Study. But, the aim of the study was not to find the optimal interval and the results do not suggest more than that the interval from injury onset to measurement should be more strictly defined.
Comment:
In the introduction I do miss a clear incentive for testing H0. I read about hamstring injuries and about TMG and the link seems to be given in this single sentence: Functional and structural muscle disorders (injuries) result in functional changes of the injured muscle
General Comment and authors Response:
Thank you for your valuable comment. The present study tested the null hypothesis of no difference in injured and non-injured BF contraction properties in elite athletes assessed by tensiomyography. The next step was to investigate the TMG classifier's highest sensitivity and specificity to separate injured and non-injured BF muscles by applying optimal cut-off values.
Comment:
But I still would ask: what is the problem and how exactly could establishing a difference in contractile properties between injured and uninjured muscle help? (I can think of some reasons, but these are not provided in the introduction.
General Comment and authors Response:
BF injuries can be functional only or structural and functional. As we know that only functional injuries cannot be detected by US or MRI alone, it is important to be able to confirm and locate them by something other than clinical examination.
Comment:
Probably some kind of file number should be added to the statement of ethical approval being provided
General Comment and authors Response:
All changes were done following the reviewer's comment. We added new sentence with the ethical codes and the information considering Ethical Institution Commision approvement for study. We added new sentence as a second paragraph at Materials and Methods chapter as follows:
The study was approved by the Commission of the Republic of Slovenia for Medical Ethics, approval study number - No. 125/03/14, and by the Institutional Ethical Board of Faculty of Sport and Physical Education, University of Belgrade Serbia, approval study number - 484-2.
Comment:
With the electrical stimulation how was current strength set and/or is the same strength used in for both legs and /or in all participants? Please add this information
General Comment and author's Response:
At the beginning of the Materials and Methods section, and at the Experimental Design chapter, we try to describe precisely how the measurement was carried out.
Comment:
I see this is provided in line 127. If the current increase is stopped when pain is evoked, doesn’t this introduce the risk that currents (and thereby muscle force and probably Dm are on average lower for the injured compared to the healthy leg? This is an important issue I would think.
General Comment and authors Response:
Measurements in which pain occurred are not part of the data and analysis.
Comment:
Were the examiners during testing aware of whether the participant was injured or not and/or in which leg the injury occurred? Please add this information, but after reading the next paragraph I think that blind assessment did not occur….turns out it did (lines 118-120) , which is a strength of the design, although I have some doubts whether this truly possible with a serious injury (as soon as the athlete enters the room you can see which leg is hurt, can’t you?)
General Comment and authors Response:
Serious problems while walking mean a high grade of injury. We have not considered these cases. We wanted to keep the procedure as unbiased as possible. I agree that it is difficult to achieve a perfect blind measurement, not only in this case. It should be added that we also routinely measured the rectus femoris and semitendinosus in the longitudinal study. The injuries were easily different and in this way, the bias was significantly reduced.
Comment:
Line 100-113 Do I understand correctly that the TMG was performed within a few days after the athletes sustained the injury? This makes my question about the introduction even more relevant, because if we can measure differences between the legs which doesn’t seem unlikely so soon after an injury) what would be next/how would this help recovery or what? If we would want to know which leg is injured, we could just ask the athlete. So you really should better explain, what maybe is obvious to you but not for the average reader, what the benefit of doing these measurements would be.
General Comment and authors Response:
In this study, we first wanted to determine whether injury (the majority of injuries were grade 0 to1-The British Athletic Classification) affects the contractile properties of the BF , which were defined through the variables Tc, Td, Dm. As we wrote in the previous comment, the signal quantifies (Tc variable) the muscle contraction changes with TMG measurement. Changes are related to a specific muscle. It is not only a matter of determining whether the leg/muscle is painful or not (this is an indicator in case there is a subjective assessment of the injury) but also a much more precise indication of the location and size of the changes on the muscles. In reality, when athletes sense changes/pain in the muscle, it may be "muscle soreness", pain in the neighboring muscle, fascia or even a problem of nerve origin. Otherwise, the changes detected on the TMG signal may take much longer (depending on the type and intensity of the injury) than 72 h, at least 2-4 weeks for grede 2 injuries, and 2-3 months for more severe injuries, grade 3 (ongoing testing). We have only made the first step with this study, which is essential for the testing full diagnostic potential of muscle twitch changes. In this study, we found that the predictive value of the Tc classification is high. Further experiments and studies could be aimed at testing whether the TMG signal can be used in the classification of rehabilitation progress and decision making in return to play challenges.
Comment:
The acronyms line 141-153 are too many for this reviewer.
fTc diff ni = percentage of the absolute difference between non-injured left and right BF Tc in the female subgroup? I will have to study line 179 to understand this…
Are all these necessary? I would stick to the main question why would one introduce left and right? Injured versus uninjured is of interest, isn’t it? Do you expect differences (for the differences between injured uninjured) between males and females?
General Comment and authors Response:
We are thankful for this comment. But, we tried to use TMG to assume that the difference between the injured and uninjured BF is statistically significantly different from the difference between the left and right BF, or between male and female. A well-known example of a significant difference between men and women is in ACL injuries incidence.
Comment:
I hope that you are not going to test for all these potential differences….this inflated your statistical power. In table one I see that many tests are performed. Did you correct for the number of t-tests performed? With so many tests and variables, you will always find a difference somewhere by chance.
General Comment and authors Response:
Yes, we used Bonferroni corrections for the between single variable p value. But, we also calculated the total number of men and women and showed the statistical differences. We want to emphasize that the primary indicator of the differences and possible classifiers is the statistics shown through the ROC/AUC analysis, not the t-test.
Comment:
Lines 179-191.., isn’t this just expressing the differences between injured and uninjured as a percentage of the mean of the maximal values of both legs ?
General Comment and author's Response:
In our study, we compare the average of the absolute differences between injured and uninjured BFs first. We then compare these differences with the average of the absolute differences between left and right BF s of uninjured athletes. This procedure is necessary because the difference between the injured and uninjured BF must be statistically significantly different(larger) from the difference of the left and right BF (in the population of uninjured athletes). Then the difference in TMG signal between injured and uninjured can be a good classifier!
So I think the answer to your question is that it is not just "expressing the differences between injured and uninjured as a percentage of the mean maximal values..."
Comment:
Line 192: Okay now I do understand that you need left and right for a negative control (still the acronyms scare me off, these do not increase the readability of your work)
General Comment and authors Response:
The authors are grateful for this comment. We are aware of the complexity of this manuscript in terms of variables and abbreviations. Therefore, we have added detailed explanations of abbreviations below all relevant tables (Table 1 and 4).
Results
Comment:
I do not think you can report the results with two significant digits (32.88 ms) this suggest an accuracy that certainly can’t be present can it? 32.9 will do for the mean values
General Comment and authors Response:
Thank you for this comment, and we accept in full this suggestion. All mean values and SD in basic descriptive data were changed (including a Table 1).
Comment:
Table 1 is quite a puzzle I see two time ni on the left side for Td Tc Dm etc. shouldn’t that be ni versus in ? This is what a meant by indicating not liking the acronyms it seems that even the authors themselves got confused….
General Comment and author's Response:
Thank you for this comment, and we accept in full this suggestion. Therefore, we have added detailed explanations of abbreviations below all relevant tables (Table 1 and 4).
Comment:
Figure 3. I am really curious about the explanation for the differences in Displacement traces around 50 ms between injured and ni. Interesting! The hump in the in-trace at 250 ms, suggests some kind of additional (reflex, or persistent inward current phenomenon) activity in the injured BF.
General Comment and authors Response:
Thank you for your valuable comment. But, this is a concrete example of one athlete. The reason for this phenomenon can hardly be understood as statistically significant, or it is not possible to say more than that it is a difference that is unique to this case. You are correct that events after 100-150ms are not necessarily associated with the primary twitch.
Comment:
Figure 4 Good figure but in the caption it reads of the female and ae subjects, so I was looking for separate female and male data, but you probably mean: of all subjects (male and female together)
I think the readability of the result section would increase by only reporting the parameters of the AUC analysis for Tc in the text (deleting lines 281-292)
General Comment and author's Response:
Figure 4 shows the absolute results of all subjects (M + F) together for a more precise visual presentation of the discriminative power of Tc in injured BF persons.
Comment:
Figure 6 and table 3 clearly show that only Tc has discriminatory potential
General Comment and author's Response:
Thank you for this comment. Therefore, Tc is recommended as the most sensitive/discriminative parameter.
Discussion
Comment:
Lines 307-340 Part of this discussion should be included in the introduction to make the objective clear.
General Comment and author's Response:
Thank you for this valuable comment. In the Introduction chapter of the manuscript, we added a new text to improve the description of the idea of this research, but we did not change the beginning of the Discussion chapter because we wanted to emphasize the theoretical setting of the research.
Comment:
This all makes sense to me, but I still am left with my main question. So great now we can objectively establish in the early stage after sustaining an injury, that there is an injury, but it is still unclear to me how this would help the rehabilitation process?
General Comment and author's Response:
Effective rehabilitation and a timely return to competition with a low probability of reinjury are the interest of every sports participant. If an injured muscle still shows a functional difference (in twitch) compared to an uninjured muscle, this indicates a risk of reinjury. If the difference decreases rapidly during rehabilitation, it means that rehabilitation is successful... Functional measurement is an additional indication of the muscle condition that can optimize individual rehabilitation and post-rehabilitation training.
Comment:
Lines 341-356 Okay fine I fully agree, but line 357: In this way, repeated assessments of large groups are possible, allowing the collection of clinically relevant longitudinal data.
General Comment and author's Response:
The idea of the sentence from line 357: In this way, repeated assessments of large groups are possible, allowing the collection of clinically relevant longitudinal data, - refers to the need to the developing system for monitoring the examined phenomenon (injured BF in athletes and the effectiveness of TMG method in detecting muscle function in terms of injury) concerning a much larger sample and in a multi-year time interval. It is assumed that the extensive data collected in this way could have more real potential for preclinical and clinical applications.
Comment:
In answer to my question about Figure 3; what causes the differences? I read lines 360-372, which is interesting. Could swelling due to oedema also play a role? If that is the case the differences in Tc between in and ni may indeed disappear relatively early in the rehabilitation period (and wouldn’t have little added value in the prognostics or treatment)
General Comment and author's Response:
Thank you for this observation, but considering our results (considering external validity), edema does not affect the classification power of Tc.
Submission Date
26 January 2022
Date of this review
11 Feb 2022 17:04:57

Round 2
Reviewer 1 Report
The article has been improved. However, it still needs to improve some details so that the quality of the article is not compromised. I thank the authors admitting the mistake of not having incorporated the conflict of interest. However, there were no need to reply me stating that "(...) so publishing one paper is no motive for compromising their professional integrity". When I made my comment about the more than clear conflict of interest (nothing happens, it is declared and that is ok) I did it in a display of transparency that must always exist in science. The job of a good reviewer is not to let anything slip, since one's prestige is also at stake. We all make a mistake when uploading an article, but these things must be made to look at. I hope that the authors take this as an attempt at transparency and improvement of the article.
1.- Title: I suggest to change it to "Tensiomyography allows to discriminate between injured and non-injured biceps femoris muscle".
2.- Abstract: Need to add information about the PRC analyses.
3.- Figure 1: Please, give more details. It should be explained at the bottom of the figure if the right and left image represents the right/left injured or non-injured thigh of the same subject or not. In addition, the figure would be more representative if the BF is colored or delimited.
4.- Page 4: Reference number 29 is not a review, change "previous reliability reviews" to "reliability studies" or replace reference 29 and add the next reference "https://www.sciencedirect.com/science/article/pii/S1050641118304176"
5.-Statistics: This is a post hoc sample size calculation, what I referred in my first report was to if simple size calculation was ad hoc performed. When post hoc sample size is used to indicate power for outcomes already observed, it is not only conceptually flawed but also analytically misleading. Please, see "Zhang, Y., Hedo, R., Rivera, A., Rull, R., Richardson, S., & Tu, X. M. (2019). Post hoc power analysis: is it an informative and meaningful analysis?. General psychiatry, 32(4)" and "Goodman, S. N., & Berlin, J. A. (1994). The use of predicted confidence intervals when planning experiments and the misuse of power when interpreting results. Annals of internal medicine, 121(3), 200-206."
You must add Monte Carlo simulation to investigate the performance of the posthoc power analysis.
6.- The first 3 paragraphs of the discussion seems like an introduction, please summarize this and adopt a correct format for discussing the results by reporting at the outset whether the results accept or reject the null hypothesis. In science you have to go more to the point, there is no need to lengthen the discussion. In addition, no reference is made to PRC subanalyses, they should be discussed in the discussion when talking about the results of the ROC curves.
Author Response
Open Review n1
(x) I would not like to sign my review report
( ) I would like to sign my review report
English language and style
( ) Extensive editing of English language and style required
( ) Moderate English changes required
(x) English language and style are fine/minor spell check required
( ) I don't feel qualified to judge about the English language and style
|
Yes |
Can be improved |
Must be improved |
Not applicable |
|
|
Does the introduction provide sufficient background and include all relevant references? |
(x) |
( ) |
( ) |
( ) |
|
Are all the cited references relevant to the research? |
( ) |
( ) |
( ) |
( ) |
|
Is the research design appropriate? |
(x) |
( ) |
( ) |
( ) |
|
Are the methods adequately described? |
(x) |
( ) |
( ) |
( ) |
|
Are the results clearly presented? |
(x) |
( ) |
( ) |
( ) |
|
Are the conclusions supported by the results? |
( ) |
(x) |
( ) |
( ) |
Comments and Suggestions for Authors
The article has been improved. However, it still needs to improve some details so that the quality of the article is not compromised. I thank the authors admitting the mistake of not having incorporated the conflict of interest. However, there were no need to reply me stating that "(...) so publishing one paper is no motive for compromising their professional integrity". When I made my comment about the more than clear conflict of interest (nothing happens, it is declared and that is ok) I did it in a display of transparency that must always exist in science. The job of a good reviewer is not to let anything slip, since one's prestige is also at stake. We all make a mistake when uploading an article, but these things must be made to look at. I hope that the authors take this as an attempt at transparency and improvement of the article.
1.- Title: I suggest to change it to "Tensiomyography allows to discriminate between injured and non-injured biceps femoris muscle".
Was changed.
2.- Abstract: Need to add information about the PRC analyses.
PCR analyses was added.
3.- Figure 1: Please, give more details. It should be explained at the bottom of the figure if the right and left image represents the right/left injured or non-injured thigh of the same subject or not. In addition, the figure would be more representative if the BF is colored or delimited.
Was added and injured BF was colored.
4.- Page 4: Reference number 29 is not a review, change "previous reliability reviews" to "reliability studies" or replace reference 29 and add the next reference https://www.sciencedirect.com/science/article/pii/S1050641118304176
Was added.
5.-Statistics: This is a post hoc sample size calculation, what I referred in my first report was to if simple size calculation was ad hoc performed. When post hoc sample size is used to indicate power for outcomes already observed, it is not only conceptually flawed but also analytically misleading. Please, see "Zhang, Y., Hedo, R., Rivera, A., Rull, R., Richardson, S., & Tu, X. M. (2019). Post hoc power analysis: is it an informative and meaningful analysis?. General psychiatry, 32(4)" and "Goodman, S. N., & Berlin, J. A. (1994). The use of predicted confidence intervals when planning experiments and the misuse of power when interpreting results. Annals of internal medicine, 121(3), 200-206."
You must add a Monte Carlo simulation to investigate the performance of the
Monte Carlo simulation (posthoc power analysis, 1000) was added.
6.- The first 3 paragraphs of the discussion seems like an introduction, please summarize this and adopt a correct format for discussing the results by reporting at the outset whether the results accept or reject the null hypothesis. In science you have to go more to the point, there is no need to lengthen the discussion. In addition, no reference is made to PRC subanalyses, they should be discussed in the discussion when talking about the results of the ROC curves.
The first 3 paragraphs of the discussion were moved to the introduction (Reviewer 2 suggestion)

Reviewer 2 Report
A lot of work was performed in manuscript and this one has improved a lot, congratulations. Below are some indications/suggestions that will result in the suggestion for publication.
Please include the authors’ initials near the emails in affiliations.
The introduction improved, but is still only three paragraphs, please consider improvement.
“BH—180.4 ± 7.6 cm; BM—70.9 ± 8.4 kg; BMI” – abbreviations should be presented in full.
Page 3 – MRI first appearance should be in full. Please correct here and confirm all the document regarding this example. In the same paragraph, for example, TMG.
Please improve all figures quality.
Page 4 – Please evaluate this line “Reprinted from the TMG Individual Report, page 3. Copyright 2010 by TMG-BMC Ltd., Ljubljana, Slovenia.”
2.5 “Statistical Analysis” is suggested.
Please indicate reference supporting the effect size.
Results should be better organized. For example, Figures 3 and 4 are introduced in page 6 but only appear in page 7 after table 1.
Please consider introducing table 2 and figure 5 separately, not in the same paragraph.
Table 3 lines are different from 1 and 2 – Please standardize.
Table 3 should have a legend and space afterward.
The fact that the study was based in sprinters (100 and 200 m) and footballers should be deeply addressed in discussion. The same for “all of the female subjects in the injured and non-injured groups were sprinters”. These facts may have influenced the results, and should be in-deep analyzed in the “discussion section”.
Please carefully review all references and the English throughout the manuscript.
Author Response
Open Review 2
(x) I would not like to sign my review report
( ) I would like to sign my review report
English language and style
( ) Extensive editing of English language and style required
(x) Moderate English changes required
( ) English language and style are fine/minor spell check required
( ) I don't feel qualified to judge about the English language and style
|
Yes |
Can be improved |
Must be improved |
Not applicable |
|
|
Does the introduction provide sufficient background and include all relevant references? |
( ) |
(x) |
( ) |
( ) |
|
Are all the cited references relevant to the research? |
( ) |
( ) |
( ) |
( ) |
|
Is the research design appropriate? |
( ) |
(x) |
( ) |
( ) |
|
Are the methods adequately described? |
( ) |
(x) |
( ) |
( ) |
|
Are the results clearly presented? |
( ) |
( ) |
(x) |
( ) |
|
Are the conclusions supported by the results? |
( ) |
(x) |
( ) |
( ) |
Comments and Suggestions for Authors
A lot of work was performed in manuscript and this one has improved a lot, congratulations. Below are some indications/suggestions that will result in the suggestion for publication.
Please include the authors’ initials near the emails in affiliations.
Was added.
The introduction improved, but is still only three paragraphs, please consider improvement.
“BH—180.4 ± 7.6 cm; BM—70.9 ± 8.4 kg; BMI” – abbreviations should be presented in full.
Was changed.
Page 3 – MRI first appearance should be in full. Please correct here and confirm all the document regarding this example. In the same paragraph, for example, TMG.
It was added and rearranged.
Please improve all figures quality.
Beter resolution curve was applied.
Page 4 – Please evaluate this line “Reprinted from the TMG Individual Report, page 3. Copyright 2010 by TMG-BMC Ltd., Ljubljana, Slovenia.”
2.5 “Statistical Analysis” is suggested.
Please indicate reference supporting the effect size.
Was done.
Results should be better organized. For example, Figures 3 and 4 are introduced in page 6 but only appear in page 7 after table 1.
Was done.
Please consider introducing table 2 and figure 5 separately, not in the same paragraph.
Was done.
Table 3 lines are different from 1 and 2 – Please standardize.
Was done.
Table 3 should have a legend and space afterward.
Was done.
The fact that the study was based in sprinters (100 and 200 m) and footballers should be deeply addressed in discussion. The same for “all of the female subjects in the injured and non-injured groups were sprinters”. These facts may have influenced the results, and should be in-deep analyzed in the “discussion section”.
It was addressed in the discussion (second part).
Please carefully review all references and the English throughout the manuscript.
It was edited by MDPI editor team.

Reviewer 4 Report
I have no further questions. The findings are clear and the presentation has been improved. It remains to be seen if these measurements will be of any help in return to play strategies.
Author Response
Thank you for your response. The usability of TMG for Return to Play strategies will be really discovered in the future, we hope.
Authors